# Specific inhibition of splicing factor activity by decoy RNA oligonucleotides

Polina Denichenko[1], Maxim Mogilevsky[1], Antoine Cléry[2], Thomas Welte[3], Jakob Biran[4], Odelia Shimshon[5], Georgina D. Barnabas[6], Miri Danan-Gotthold[7], Saran Kumar [8], Eylon Yavin[5], Erez Y. Levanon [7], Frédéric H. Allain[2], Tamar Geiger[6], Gil Levkowitz [4] & Rotem Karni [1]

Alternative splicing, a fundamental step in gene expression, is deregulated in many diseases. Splicing factors (SFs), which regulate this process, are up- or down regulated or mutated in several diseases including cancer. To date, there are no inhibitors that directly inhibit the activity of SFs. We designed decoy oligonucleotides, composed of several repeats of a RNA motif, which is recognized by a single SF. Here we show that decoy oligonucleotides targeting splicing factors RBFOX1/2, SRSF1 and PTBP1, can specifically bind to their respective SFs and inhibit their splicing and biological activities both in vitro and in vivo. These decoy oligonucleotides present an approach to specifically downregulate SF activity in conditions where SFs are either up-regulated or hyperactive.

[1] Department of Biochemistry and Molecular Biology, Institute for Medical Research Israel-Canada, Hebrew University-Hadassah Medical School, Jerusalem 9112001, Israel. [2] Institute of Molecular Biology and Biophysics, ETH Zurich, Hönggerbergring 64, 8093 Zurich, Switzerland. [3] Dynamic Biosensors, GmbH, Lochhamer Strasse 15, 82152 Martinsried/Planegg, Germany. [4] Department of Molecular Cell Biology, Weizmann Institute of Science, Rehovot 76100, Israel. [5] Department of Medicinal Chemistry, Institute for Drug Research, Hebrew University-Hadassah Medical School, Jerusalem 9112001, Israel. [6] Department of Human Molecular Genetics and Biochemistry, Sackler Faculty of Medicine, Tel Aviv University, Tel Aviv 6997801, Israel. [7] Mina and Everard Goodman Faculty of Life Sciences, Bar-Ilan University, Ramat Gan 52900, Israel. [8] Department of Developmental Biology and Cancer Research, Institute for Medical Research Israel-Canada, Faculty of Medicine, Hebrew University-Hadassah Medical School, Jerusalem 9112001, Israel. Correspondence and requests for materials should be addressed to R.K. (email: rotemka@ekmd.huji.ac.il)

The involvement of SFs in multiple diseases and processes was the driving force behind the idea to develop SF specific inhibitors. SFs bind, in most cases, to a degenerate motif in the pre-mRNA of the target gene and either recruit or repel the spliceosome to/from nearby splice sites[1]. Many SFs also possess RNA-independent functions, such as protein–protein interactions in cellular complexes, which are essential for proper cellular functions[2]. Inhibition of SF expression by siRNAs or antisense oligonucleotides could have broad detrimental effects on cell fate[3–5]. Therefore, development of an efficient SF inhibitor should ideally target only the splicing activity of the factor, without interfering with its other activities.

Current oligonucleotides-based technologies include: Antisense GAPmers, which are designed to knockdown gene expression by binding to specific mRNAs and activating their degradation by RNAse H ; Splice Switching Oligos, which hybridize to pre-mRNA molecules, interfere with the binding of splicing factors or spliceosomal components and shift the splicing between splice sites or affect inclusion/skipping of specific exons; and siRNAs, which are designed to knockdown gene expression and are usually double stranded (reviewed in the ref. [6]). All of these oligonucleotide technologies are based on binding/hybridization to either pre-mRNA or mRNA. Here we present a technology using sense oligonucleotides that bind to RNA binding proteins rather than RNA. The only known similar approaches of nucleic acids designed to bind proteins are DNA (double stranded) oligonucleotides, that act as transcription factor decoys[7,8] and RNA aptamers, which are RNA molecules (sometimes much longer than the RNA oligonucleotides mentioned above) with a specific 3D structure that can bind many types of proteins according to the designed specificity (not only RNA binding proteins)[9].

In order to test the feasibility of using decoy RNA oligonucleotides to inhibit splicing factor activity we chose to target three alternative splicing factors; RBFOX1/2, PTBP1, and SRSF1. RBFOX1 and RBFOX2 are members of a splicing factor family known to be involved in multiple diseases. Altered expression of RBFOX2 in ovarian and breast cancer causes altered splicing of specific targets[10]. RBFOX1, also known as A2BP1, is deleted in 10% of glioblastoma multiforme and can act as a tumor suppressor[11]. Abnormal expression of RBFOX1/2 may play a role in neuroblastomas and glioblastomas[12–15], epilepsy and mental retardation[16]. These proteins have also been shown to be involved in heart and muscle development and function in zebrafish muscle development[17]. Recent studies showed that RBFOX2 is important for myoblast fusion during myogenesis in mice[18] and that repression of RBFOX2 is linked to heart failure[19]. RBFOX1/2 regulate hundreds of splicing events as measured by their direct RNA binding using RNA CLIP and RNA-seq experiments[20–22]. PTBP1 and SRSF1 are two splicing factors known to be involved in cancer. PTBP1 is aberrantly elevated in glioblastomas and serves as a marker for glioblastoma progression[23]. The oncoprotein splicing factor SRSF1, is amplified in breast cancer and is a potential target for cancer therapy[24–26]. Most splicing factors bind degenerate motifs in the pre-mRNA of their target genes. However, RBFOX1/2 proteins bind a well-defined motif (U)GCAUG[12,16], which makes it an ideal target for specific inhibition.

A requirement for any SF inhibitor is that its activity should be limited to targeting only the splicing activity of the factor, without interfering with other activities of the factor. Many splicing factors possess RNA-independent functions. These functions include protein–protein interactions in cellular complexes, which are essential for proper cellular functions. For example, RBFOX proteins are part of a large assembly of splicing regulators (LASR). One of the LASR subunits, hnRNP M, also has a function independent of the complex and RBFOX1/2 can stimulate splicing repression mediated by hnRNP M in an RNA binding-independent manner[21]. Another example of a splicing factor functioning in protein interactions is the oncoprotein SRSF1. SRSF1 has a role in stabilizing p53 through interaction with RPL5, which induces senescence in cells[2]. Therefore, simple knock-down or inhibition of the SF would have multiple negative effects on the cell. Ideally, a decoy oligonucleotide should target only the splicing activity of the factor. Moreover, the efficacy of splicing factor knockdown by siRNAs or antisense oligonucleotides is dependent on the half-life of the factor and the integrity of cellular components (e.g., RNAi machinery), while the decoy oligonucleotide can act directly and immediately, independent of any cellular component.

Here we provide a proof-of-principle demonstration that RNA decoy oligonucleotides targeting splicing factors RBFOX1/2, SRSF1, and PTBP1, can specifically bind to their respective SFs and inhibit their splicing and biological activities both in vitro and in vivo. Decoy oligonucleotides present a methodology to specifically downregulate SF activity and have the potential to treat diseases where SFs are upregulated. Relevant to this, we demonstrate that inhibition of the oncogenic splicing factors SRSF1 and PTBP1 inhibits the oncogenic properties of cancer cells.

## Results

**Design of decoy oligonucleotides**. We designed RNA decoy oligonucleotides directed against three alternative SFs; RBFOX1/2 (RBFOXi), PTBP1 (PTBP1i) and SRSF1 (SF2i1 and SF2i2). The oligonucleotides were predicted to bind the RRM domains of each SF (Fig. 1a), based on several studies[1,12,27,28]. Each oligonucleotide contains three/four tandem motif repeats. An oligonucleotide that does not resemble any of the known exonic splicing enhancer motifs[29], scrambled (SCRM), was used as a control in all experiments. Several biological experiments were performed with additional control oligonucleotides. The decoy oligonucleotides are single stranded RNA molecules with a 2′-O-methyl modification on the ribose of each nucleotide, which increases the stability of the molecule[6]. For some in vitro experiments these oligonucleotides were used, with an additional Cy5 or biotin modification on the 5′ end. For in vivo experiments in zebra fish, oligonucleotides were also modified with a phosphorothioate backbone.

**RBFOX1/2 decoys bind specifically and affect splicing**. Transfection of the Cy5 labeled RBFOXi oligonucleotides into U87MG cells resulted in nuclear localization (Supplementary Fig. 1A). Cy5 RBFOXi oligonucleotides were detected in the nucleus and cytoplasm as early as 1 h post-transfection but were not detected in the endosomes (Supplementary Fig. 2). The amount of oligonucleotides in the cytoplasm and the nucleus increased at 3 h post-transfection (Supplementary Fig. 3A). Even after 11 h post-transfection there was no evidence of co-localization of the oligonucleotides with endosomes (Supplementary Fig. 3B, Supplementary movie 1, 2, 3). Performing quantification on 213 U87MG cells transfected with RBFOXi we estimated that at least 23% ±10 of the oligonucleotides are localized to the nucleus (Supplementary Data 1).

A pull-down assay performed with nuclear extract efficiently pulled-down RBFOX2 protein only with the biotinylated RBFOXi oligonucleotide and not with the SCRM oligonucleotide. In addition, RBFOXi did not interact with other splicing factors that were examined as controls (Fig. 1b). Mass spectrometry-based proteomic analysis of the pulled-down proteins revealed specific binding of only RBFOX1/2 to the RBFOXi oligonucleotide (Fig. 1c and Supplementary Data 2). NMR is a very sensitive

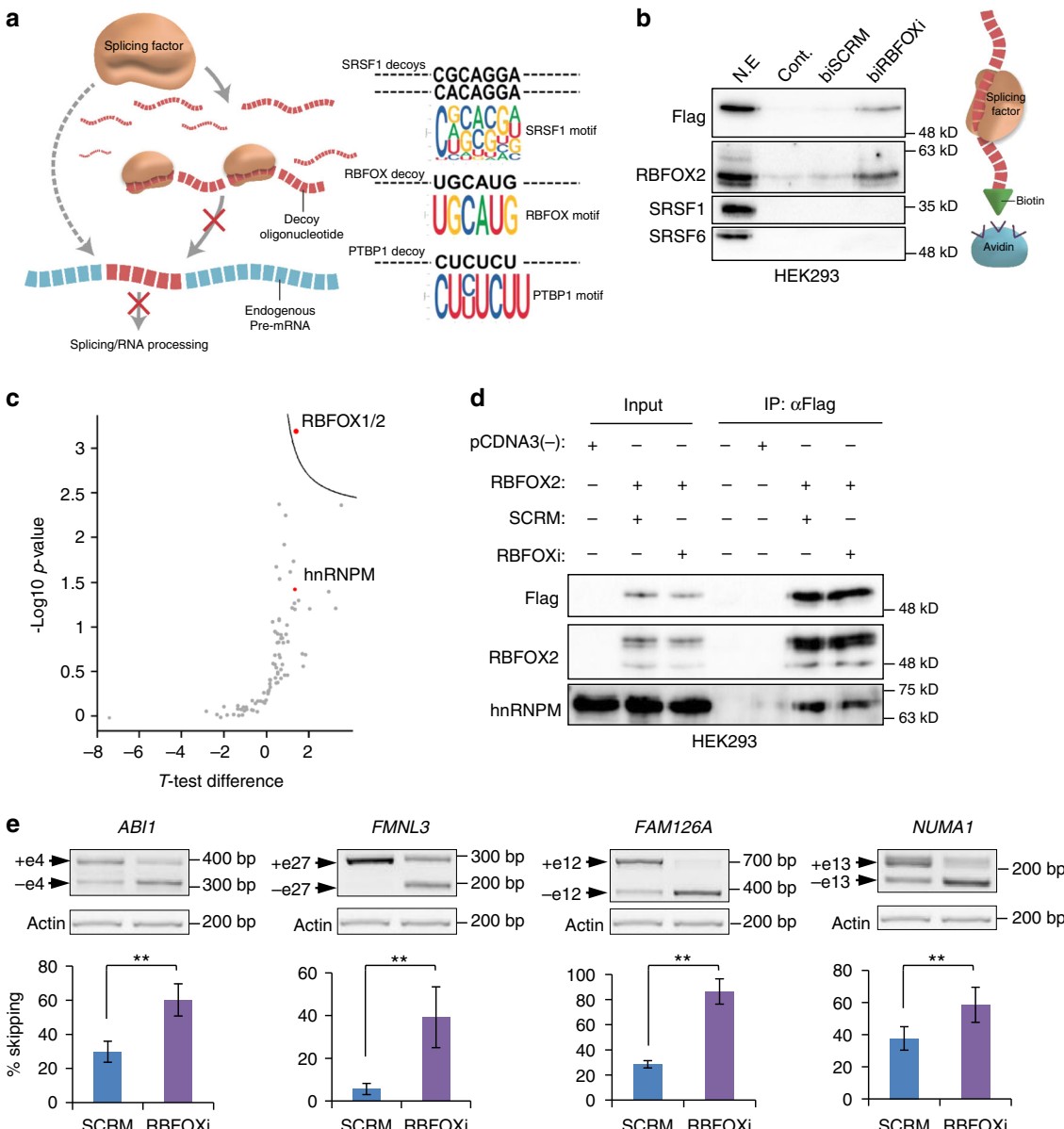

**Fig. 1** RBFOX1/2 decoy oligonucleotide binds RBFOX2 and affects its splicing targets. **a** Scheme representing inhibition of RNA binding proteins by decoy oligonucleotides (left panel). Motif sequences of decoy oligonucleotides (black letters) compared to consensus motif sequences of specific splicing factors (colored sequences) (right panel). **b** Western blot of proteins pulled down with biotin conjugated RBFOXi oligonucleotide (biRBFOXi) or SCRM (biSCRM) using nuclear extracts from HEK293 cells overexpressing RBFOX2. **c** Volcano plot showing statistically significant proteins pulled down with biotin conjugated RBFOXi, compared to SCRM control, using nuclear extracts from HEK293 overexpressing RBFOX2. **d** Western blot of HEK293 cells co-transfected with pcDNA3 Flag RBFOX2 and either RBFOXi or SCRM. Cells were fractionated and nuclear extract lysates were used for immunoprecipitation using anti-Flag antibody. Input and IPs were analysed by immunoblotting with the indicated antibodies. **e** RT-PCR and quantification of known RBFOX1/2 targets in U87MG cells transfected with 2.5 μM of either SCRM or RBFOXi. Gel image of representative experiment is shown above each graph. The same actin control is shown for all panels, as the gels are from the same experiment. \*\**p*-value <0.003, calculated using Student's *t*-test (two-tailed). Data represent means ± SD of six independent biological samples

approach allowing the detection of weak interactions. We used NMR to compare RBFOX1 RRM (identical to RBFOX2 RRM) interaction with the RBFOXi oligonucleotide compared to a control oligonucleotide (UCAGAGGA), which was shown to be specifically recognized by SRSF1[25]. Only small chemical shifts were observed with the control oligonucleotide compared to the very large perturbations observed with the RBFOXi oligonucleotide (Supplementary Fig. 4). This result further demonstrates the binding specificity of the RBFOX1/2 binding motif and suggests that the SRSF1 motif would not efficiently interact with RBFOX1/2 proteins.

A recent study showed that RBFOX2 is in a complex with hnRNP M, in an RNA independent manner[21]. Indeed, in the mass-spectrometry analysis of the pull-down assay we identified hnRNP M as a RBFOXi interactor, even though it was not statistically significant (Supplementary Data 2). A co-IP assay showed that the protein–protein interaction between RBFOX2 and hnRNP M remained intact even after transfection of RBFOXi into the cells, re-enforcing the prediction that the decoy oligonucleotide does not interfere with protein–protein interactions (Fig. 1d). In addition, introduction of RBFOXi into cells did not significantly affect the protein levels of RBFOX2 (Supplementary Fig. 1B).

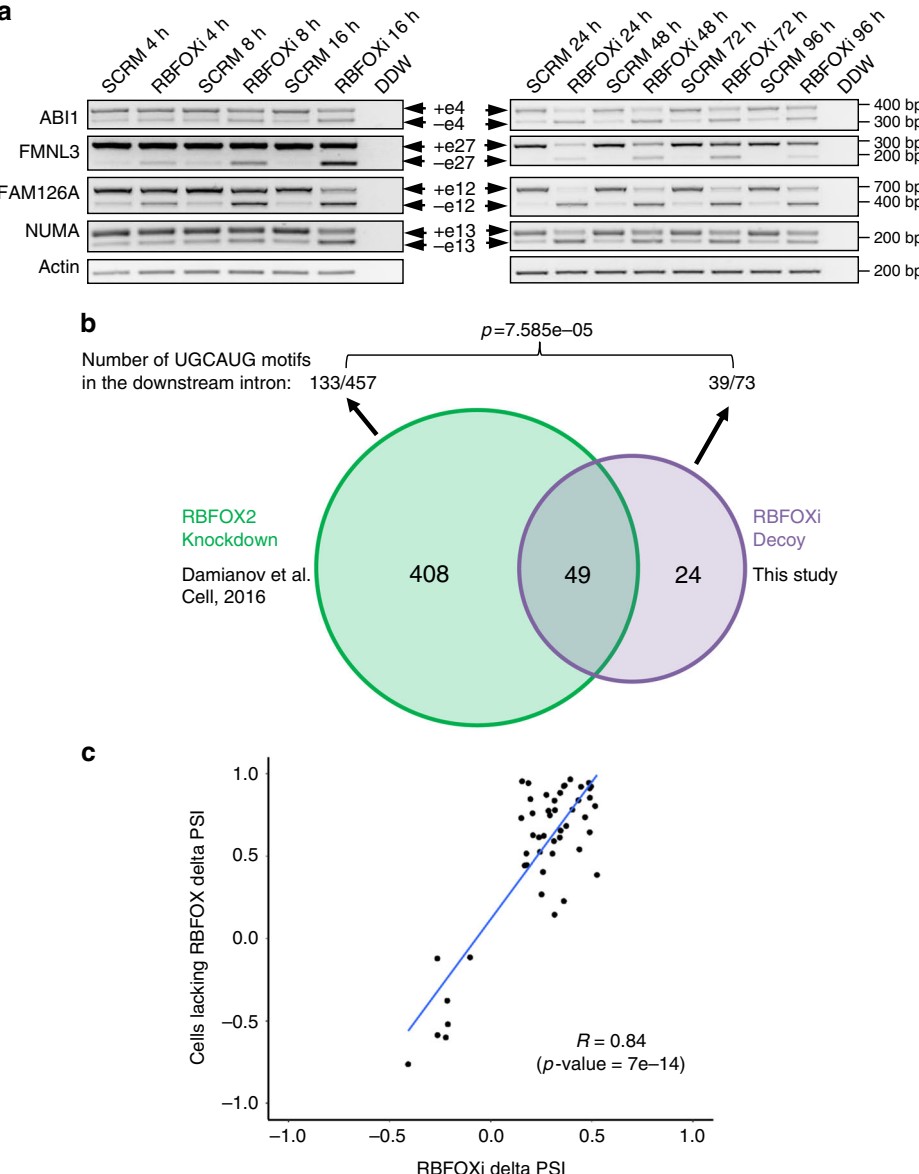

**Fig. 2** RNA-seq of RBFOXi transfected U87MG cells. **a** RT-PCR of RBFOX1/2 targets in U87MG cells transfected with 2.5 μM of either RBFOXi or SCRM harvested at the indicated time points. **b** Venn diagram showing the overlap of exons identified as having altered splicing in RBFOX knockdown cells (green) (as previously reported by Damianov et al. [21]) and our RBFOXi treated cells (purple). Number of FRBOX1/2 motifs (UGCAUG) found in the introns downstream of the alternatively spliced exons in each study are shown above the Venn diagram. **c** Scatter plot showing the correlation of change in splicing (delta PSI) of the 49 significantly alternatively spliced exons detected in both studies (**b**) (FDR < 0.05)

After determining that RBFOXi successfully binds to RBFOX2 (Fig. 1b) we examined if this binding results in changes in splicing of known RBFOX1/2 targets[16,22]. Splicing of known target genes was affected by introduction of RBFOXi to U87MG cells (Fig. 1e) and MDA-MB-435S cells (Supplementary Fig. 1C) in a dose dependent manner (Supplementary Fig. 1D). Splicing changes of some of the targets were observed as early as 4 h after transfection and persisted up to 96 h, demonstrating the rapid and enduring biological effect of the decoy oligonucleotides (Fig. 2a). These alternative splicing changes are similar to the changes observed in cells upon RBFOX2 knockdown (Supplementary Figure 1E-F) and previous reports[16,22]. In order to assess the effect of RBFOXi across the entire transcriptome we performed RNA-seq on U87MG cells transfected with either RBFOXi or SCRM. Data analysis with rMATS[30] revealed splicing changes (between RBFOXi and SCRM transfected cells) in 73 alternative exons

(Table S3); 49 of which were previously reported to have altered splicing in RBFOX1/2 knockdown cells (changes were in the same direction in both studies)[21] (Fig. 2b, c and Supplementary Data 3). Moreover, out of the 73 splicing events affected by RBFOXi, 39 of the downstream introns contained the consensus RBFOX1/2 binding motif UGCAUG, while out of the 457 splicing events affected by RBFOX2 knockdown only 133 of the downstream introns contained this motif. These differences are statistically significant (Fig. 2b). The fact that RBFOX1/2 can also affect splicing in an RNA binding-independent manner[21] might explain why the decoy oligonucleotide affected the splicing of fewer genes than RBFOX1/2 knockdown.

**Increased number of binding sites increases affinity**. We used the switchSENSE approach[31,32] (Fig. 3a) to investigate whether an

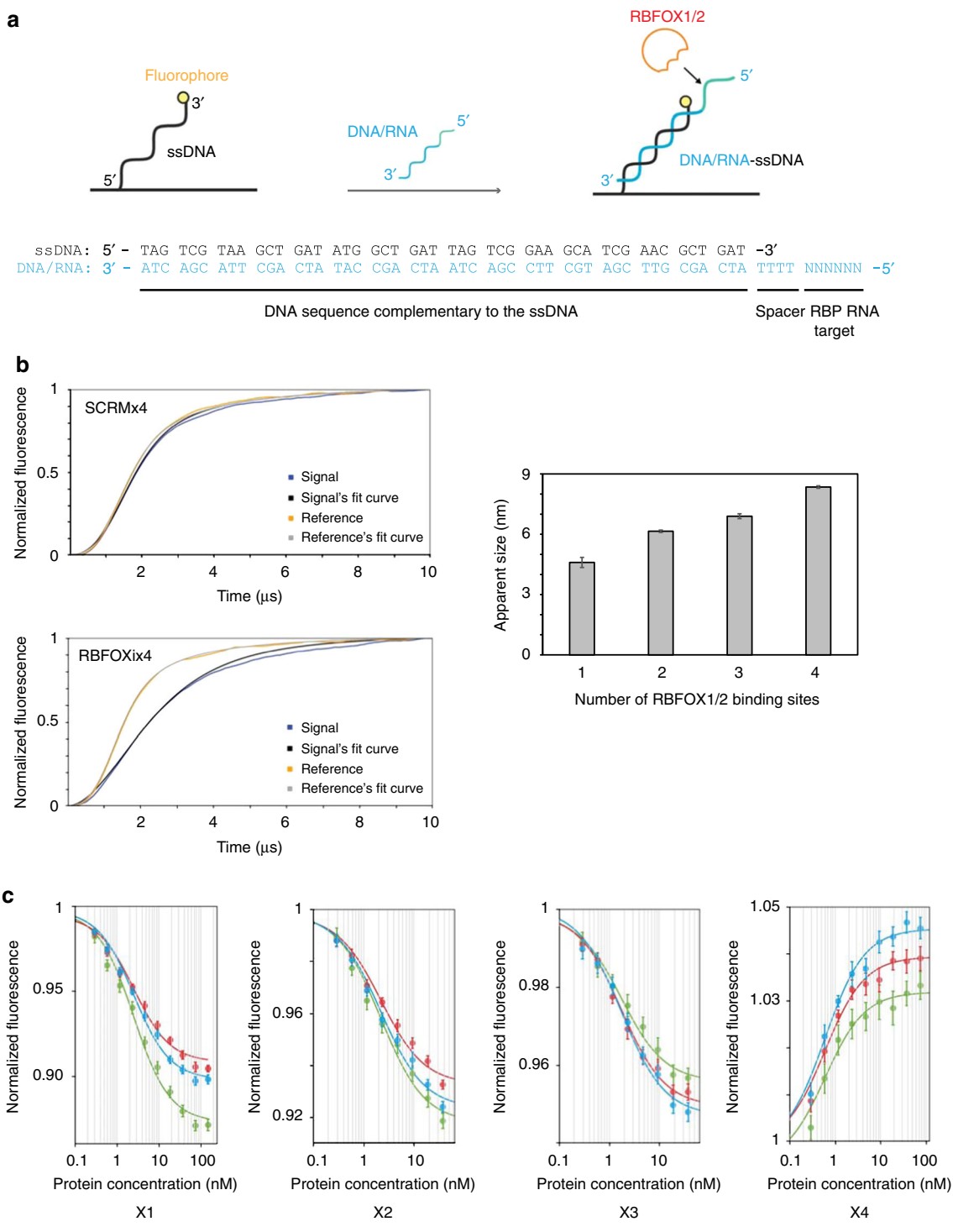

**Fig. 3** RBFOX1 RRM interaction with RNAs containing consecutive binding sites using switchSENSE. **a** Schematic view of hybridization of the DNA/RNA chimeric molecule to the ssDNA covalently attached to the biochip electrodes. The sequences of ssDNA and DNA/RNA molecules are shown. **b** Sizing experiments performed with 150 nM of RBFOX1 RRM in T40 buffer at 37 °C. Results are shown for the RBFOX1×4 and SCRM×4 DNA/RNA molecules (left side) and a graph shows the apparent size measured with RNA containing one (×1), two (×2), three (×3), and four (×4) consecutive RBFOX1 binding sites (right side). **c** Titrations performed using switchSENSE on three different electrodes (red, blue and green colors) with increasing concentrations (0.3, 0.6, 1.2, 2.3, 4.7, 9.4, 18.8, 37.5, 75, 150 nM) of RBFOX1 RRM and ×1, ×2, ×3, and ×4 DNA/RNA molecules. Experiments were performed at 37 °C in T100 buffer. $K_D$ values obtained from global Langmuir fits are indicated for each interaction with the corresponding standard deviation

oligonucleotide containing multiple tandem motifs would have increased affinity for the RBFOX1/2 RRM. Oligonucleotide sequences containing either, one, two, three or four consecutive binding sites (RBFOXi ×1 to ×4) and random sequences of

similar lengths as controls (SCRM×1 to ×4) were first tested for protein binding kinetics. Increased binding caused delay in the apparent fluorescence kinetics (Fig. 3b). We found that the RBFOX1 RRM binds oligonucleotides containing the RBFOX1/2

motif with delayed kinetics, which increased with increasing number of binding sites (Fig. 3b bottom left panel, Supplementary Fig. 5). No binding was detected with the scrambled oligo (Fig. 3b, top left panel, Supplementary Fig. 5). Although the model used to evaluate the data does not allow exact determination of the molecular weight and stoichiometry of protein-RNA complexes, the method does indicate changes in the size of the protein-RNA complexes. Whereas no binding was observed with the scrambled sequences, an increase in size was detected with oligonucleotides with increasing number of binding sites (Fig. 3b, right panel). We then investigated whether an increase in the number of RBFOX1/2 motifs was associated with increased affinity to RBFOX1/2 RRM. We performed switchSENSE titrations of RBFOXi ×1 to ×4 oligonucleotides with increasing concentrations of RBFOX1/2 RRM to determine the Kd values. We observed a positive correlation of affinity of RBFOX1/2 RRMs with increased number of binding sites. While the oligonucleotide with one motif repeat had a Kd of 2.5 nM, the oligonucleotide containing four repeats had a Kd of 0.6 nM (Fig. 3c). Given that we observed an increased number of RBFOX1/2 RRMs bound to the oligonucleotides with multiple motifs (Fig. 3b), we hypothesize that the lower Kd of the oligonucleotide containing four repeats is due to cooperativity in binding of the proteins. In agreement with these results, oligonucleotides containing four or five binding sites had the strongest effect on splicing (Supplementary Fig. 7B).

**RBFOX1/2 decoy affects splicing and muscle development.** Gallagher et al.[17] have shown that RBFOX1/2 proteins are essential for the development and function of muscle and heart in zebrafish. Likewise, Singh et al. showed that RBFOX1/2 is essential for muscle development in mammals[18]. We utilized the zebrafish model system to determine if inhibition of RBFOX1/2 proteins with RBFOXi results in a similar phenotype as RBFOX knockdown by morpholino antisense oligonucleotides (which block the expression of RBFOX1/2). Fertilized zebrafish eggs were injected with either RBFOXi or SCRM oligonucleotides (Fig. 4a). We observed changes in alternative splicing of four muscle related RBFOX targets[17] in RNA isolated from the larvae derived from the RBFOXi injected fertilized eggs (Fig. 4b). Phalloidin labeling of F-actin filaments of these larvae revealed disorganized and wavy muscle fibers, with a higher dose of RBFOXi (8pg) having a more severe phenotype (Fig. 4c, d). This phenotype is similar to those reported by Gallagher et al. upon RBFOX1/2 knockdown[17], demonstrating that inhibition of the RNA binding properties of RBFOX1/2 proteins by the decoy oligonucleotides and the concomitant change in alternative splicing, result in a delayed muscle development phenotype in vivo.

**PTBP1 decoy affects splicing and inhibits oncogenic properties.** To examine the possible therapeutic potential of decoy oligonucleotides against splicing factors, we designed a decoy oligonucleotide against another splicing factor, PTBP1, which is upregulated in several cancers and promotes pro-oncogenic splicing events[23,33–35]. The decoy oligonucleotide against PTBP1 (PTBP1i) is composed of four repeats of the PTBP1 binding motif, CUCUCU[36,37]. Pull-down assays using HEK293 nuclear extracts showed specific binding of PTBP1 to biotinylated PTBP1i and not to the SCRM oligonucleotide. Similar to RBFOXi, PTBP1i did not interact with other splicing factors (Fig. 5a). Mass spectrometry-based proteomic analysis of the PTBP1i pulled-down fraction showed strong and significant binding to PTBP1 (Fig. 5b and Supplementary Data 4) with no down regulation of PTBP1 protein levels in cells transfected with PTBP1i (Supplementary Fig. 6A). Furthermore, we observed splicing changes of

known PTBP1 targets, PKM2[33,35], RTN4[38], SNAP91, and PTBP2[39], when PTBP1i was introduced into breast cancer cells (Fig. 5c). The known involvement of PTBP1 in cancer motivated us to check if inhibition of its splicing activity could inhibit transformation. Indeed, transfection of breast cancer and glioblastoma cells lines with PTBP1i reduced proliferation (almost two-fold), inhibited the formation of colonies in a clonogenic assay and inhibited colony formation in soft agar (Fig. 5d–f and Supplementary Fig. 6B-C). These results expand the possibility of use of decoy oligonucleotides to inhibit another splicing factor, PTBP1, and demonstrate that inhibition of PTBP1 with a decoy oligonucleotide can inhibit oncogenic properties of breast and glioblastoma cancer cells.

**SRSF1 decoys bind SRSF1 and affect splicing.** Two decoy oligonucleotides, differing by a single nucleotide in the motif sequence, were designed against SRSF1; SF2i1 (5′ (CACAGGA)n, $n = 3$ repeats) and SF2i2 (5′ (CGCAGGA)n, $n = 3$ repeats) (Fig. 1a, Supplementary Table 1). A pull-down assay performed with nuclear extracts showed efficient pull-down of SRSF1 protein with the biotinylated SF2i2 oligonucleotide and not with the SCRM oligonucleotide. In addition, SF2i2 did not interact with other splicing factors that were examined as controls (Fig. 6a). Mass spectrometry-based proteomic analysis of the pulled-down proteins revealed specific binding of SRSF1 to the SF2i2 oligonucleotide. However, many additional proteins were also pulled down in the assay (Supplementary Fig. 7A and Supplementary Data 5). Many of these proteins were previously shown to bind SRSF1 directly in an RNA-independent manner[40], thus suggesting that they interact directly with SRSF1 protein and not with the decoy oligonucleotide. In addition to its role in alternative splicing, SRSF1 also functions in an RNA-independent manner, as part of the RPL5-MDM2 complex[2]. We examined if the interaction between SRSF1 and RPL5 still persists in cells transfected with SF2i2. We found no interference of this interaction in cells transfected with SF2i2 (Fig. 6b). This confirms that the decoy oligonucleotide does not interfere with protein-protein interactions in cellular complexes, which can be essential for cellular functions. Transfection of the two decoy oligonucleotides (SF2i1 and SF2i2) into U87MG and MDA-MB-231 cells resulted in changes in splicing of four known targets of SRSF1, INSR[41], U2AF1[42], MKNK2[24,43], and USP8[25] with SF2i2 having a stronger effect than SF2i1 (Fig. 6c and Supplementary Fig. 7C). In an attempt to calibrate the optimal number of motif repeats necessary for the splicing effect, we tested SF2i2 oligonucleotides containing one, two, three, four, or five repeats of the CGCAGGA motif. The effect on splicing of SRSF1 target genes positively correlated with the number of motif repeats (Supplementary Fig. 7B). These results are consistent with the results obtained for RBFOXi binding to RBFOX1 RRM using the switchSENSE approach, where we observed higher affinity for RBFOXi ×4 compared to RBFOXi ×1 (Fig. 3c and Supplementary Fig. 5). Splicing patterns of SRSF1 splicing targets were altered in a dose dependent manner (Supplementary Fig. 7D) and without any effect on SRSF1 protein levels (Supplementary Fig. 7E).

To further evaluate the specificity of all decoy oligonucleotides, two splicing targets of each splicing factor were chosen; FMNL3 and NUMA1 for RBFOX1/2, RTN4, and SNAP91 for PTBP1 and USP8 and U2AF1 for SRSF1. Changes in splicing of each target were evaluated after introduction of each decoy oligonucleotide (Supplementary Fig. 8). Changes in splicing of FMNL3 and NUMA1 were seen only after transfection of RBFOXi but not PTBP1i or SF2i2 (Supplementary Fig. 8A). Changes in splicing of RTN4 and SNAP91 were seen only after transfection with PTBP1i (Supplementary Fig. 8B) and significant changes in USP8 and

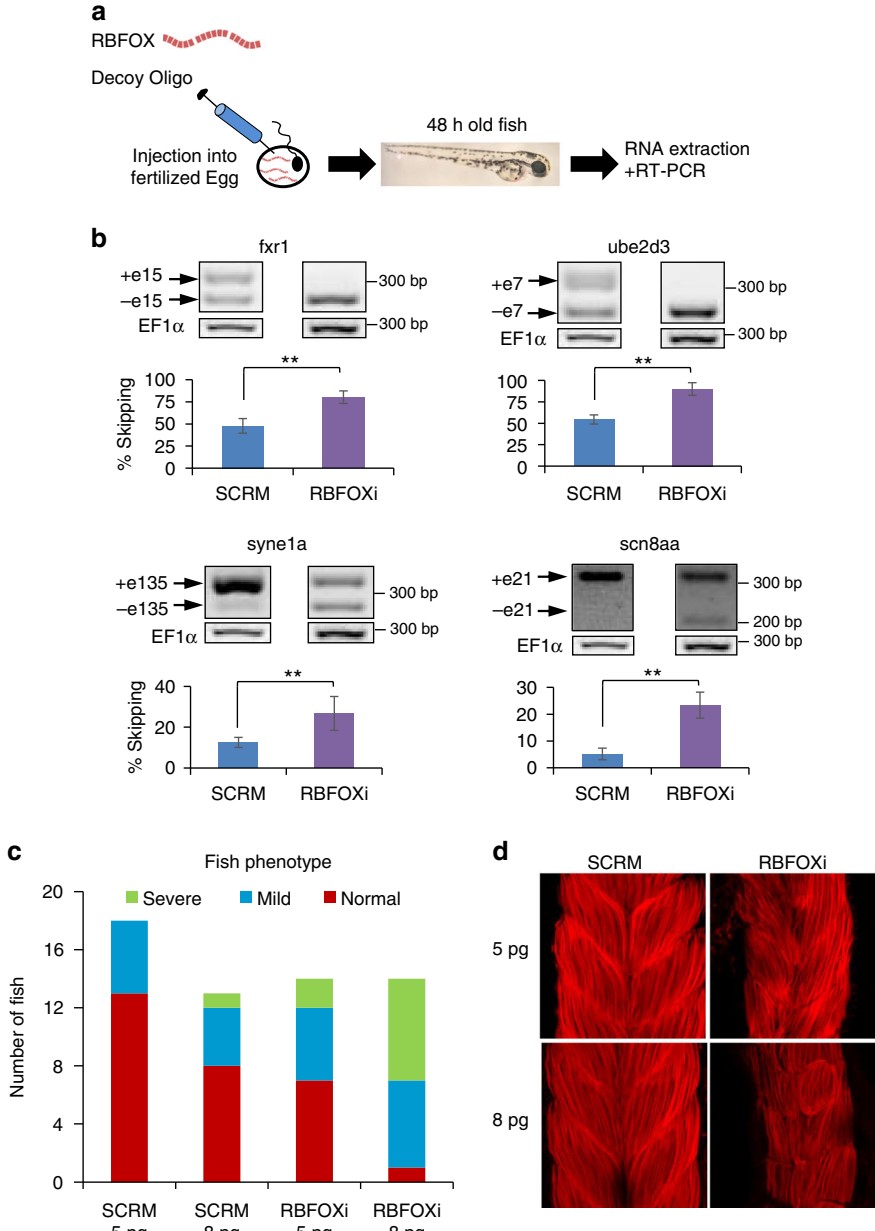

**Fig. 4** RBFOXi affects alternative splicing and muscle development in zebrafish. **a** Scheme of experimental design, where either 5pg or 8pg of RBFOXi (or SCRM) were injected to fertilized zebrafish eggs. **b** RT-PCR of known splicing targets of RBFOX1/2 in 48 h old larva, after injection with 8pg of either SCRM or RBFOXi. RNA was extracted and reverse transcribed from each individual larva. Gel image of a representative experiment is shown above each column. The same EF1α control is shown for all panels, as the gels are from the same experiment **$p$-value < 0.003, calculated using Student's $t$-test (two-tailed). Data represent means ± SD of ten zebrafish samples. **c** Quantification of phenotype severity 48 h post fertilization after injection with either SCRM or RBFOXi (5pg, 8pg). Phenotypes were assigned by visualization of fiber formation and segmental structures. **d** Pictures of phalloidin staining of F-actin fibers in fish tails 48 h post fertilization after injection with 8pg of either SCRM or RBFOXi

*U2AF1* splicing were observed only after transfection with SF2i2 but not with any of the other decoy oligonucleotides (Supplementary Fig. 8C). In addition, we used three unrelated control oligonucleotides; Cont.1, Cont.2, Cont.3. These control oligonucleotides were designed as antisense oligonucleotides against exon 78 of the dystrophin (DMD) pre-mRNA which is expressed mostly in muscle tissues. Moreover, these antisense oligonucleotides did not affect DMD exon 78 splicing, which has no known effect on tumorigenic properties or splicing. We examined the effect of these control oligonucleotides on splicing of SRSF1 targets; *USP8* and *U2AF1*. There were no detectable splicing changes in these targets after introduction of the control

oligonucleotides (Supplementary Fig. 9A). These results further strengthen our conclusion that each decoy oligonucleotide acts specifically on its targeted splicing factor. It is important to note that other splicing events may not behave in a similar manner to these specific splicing targets, since they might contain different cis-elements and be regulated by several different splicing factors simultaneously.

**SRSF1 decoys activate the p38-MAPK pathway and inhibit NMD.** One of the known splicing targets of SRSF1 is the kinase *MKNK2*[43]. Elevated expression of *MNK2a*, a result of alternative

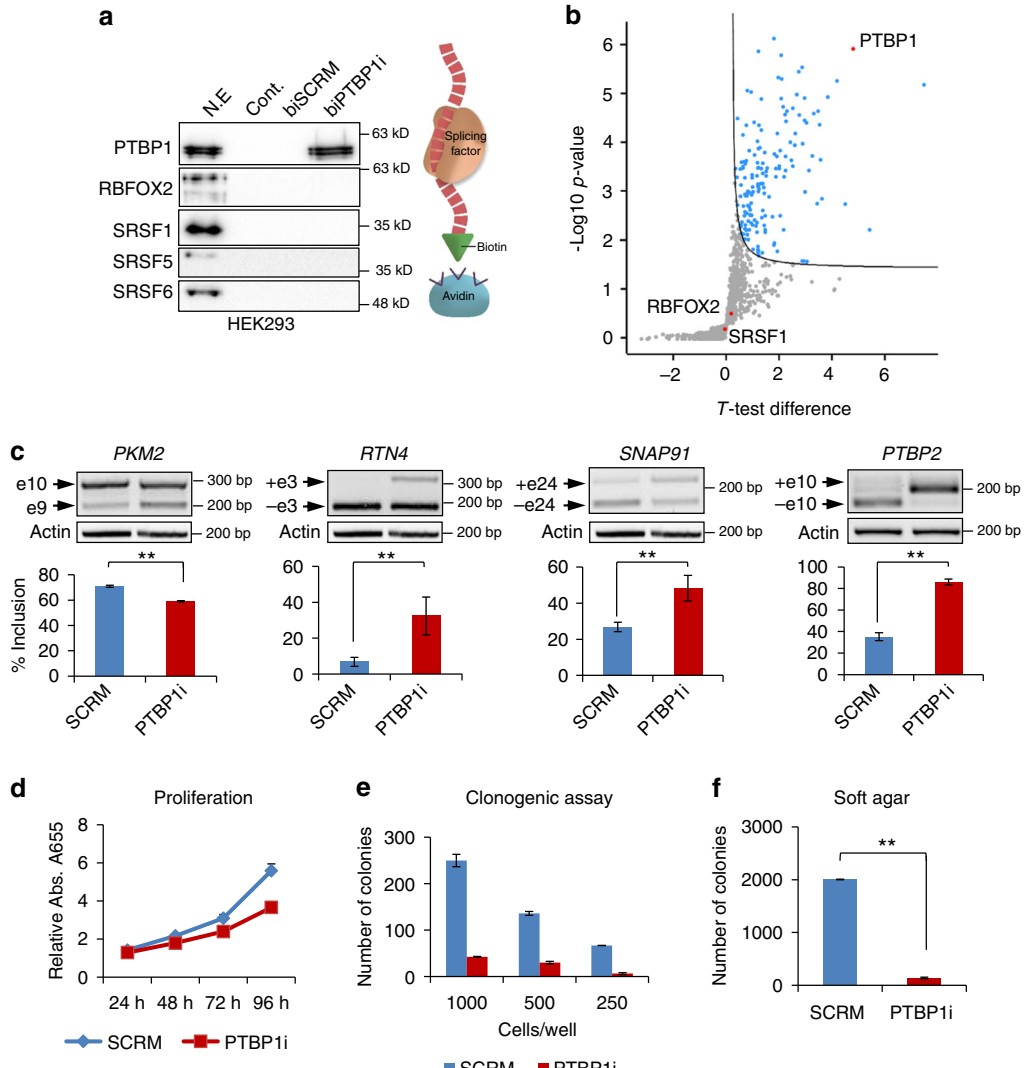

**Fig. 5** PTBP1 decoy oligonucleotides inhibit its splicing and biological activities in breast cancer cells. **a** Western blot of proteins pulled down with biotin conjugated PTBP1i (biPTBP1i) or SCRM (biSCRM) using HEK293 nuclear extracts. **b** Volcano plot showing statistically significant proteins pulled down with biotin conjugated PTBP1i, compared to SCRM control, using HEK293 nuclear extracts. **c** RT-PCR and quantification of known PTBP1 splicing targets in MDA-MB-231 cells 48 h after transfection with either 5 μM biSCRM or biPTBP1 oligonucleotides. **p-value <0.0008. Data represent means ± SD of five independent biological samples. **d** Proliferation assay of MDA-MB-231 cells transfected as in **c**. Four hours after transfection 4000 cells/well were seeded in six replicates. **p-value <0.0002 for all time points. Data represent means ± SD of six replicates. **e** Clonogenic assay of MDA-MB-231 cells transfected as in **c** plated at different densities (250, 500, or 1000 cells/well). n = 2. Data represent means ± SD of duplicate wells. **f** Soft agar colony growth assay on MDA-MB-231 cells transfected as in **c**. Graphs represent quantification of 10 fields counted in duplicate (total of 20 fields). **p-value <1.7E−27. Data represent means ± SD of two wells. p-values were calculated using Student's t-test (two-tailed)

splicing of *MKNK2*, is known to activate the p38-MAPK stress pathway[43]. We examined if inhibition of SRSF1 by the decoy oligonucleotides activates this stress pathway. We observed increased phosphorylation of p38-MAPK and its substrate ATF2[44] upon transfection with either SF2i1 or SF2i2 (Fig. 6d and Supplementary Fig. 7F) and up-regulation of expression of downstream transcriptional targets of the p38-MAPK pathway; *COX2*, *c-FOS*, and *IL-6*[43,45] (Fig. 6e and Supplementary Fig. 7G). Activation of the p38-MAPK pathway was specific to SRSF1 decoy oligonucleotides and was not observed after transfection with RBFOXi and PTBP1i or with the additional control oligonucleotides Cont.1-Cont.3 (Supplementary Fig. 9B). The role of SRSF1 in activating the nonsense mediated decay (NMD) pathway is also mediated through its RRM domains[46,47]. We therefore predicted that inhibition of SRSF1 with decoy oligonucleotides would inhibit NMD. We observed an accumulation of three

known endogenous NMD-prone transcripts[48,49] after treatment with the SRSF1 decoy oligonucleotides, suggesting that NMD is inhibited (Fig. 6f and Supplementary Fig. 7H). Once again, in agreement with previous results (Fig. 6c–e and Supplementary Fig. 7G), SF2i2 had a stronger effect than SF2i1. These results demonstrate that the SRSF1 decoy oligonucleotides inhibit SRSF1's ability to enhance NMD, to regulate alternative splicing, and lead to activation of the p38-MAPK stress pathway without interfering with non-RNA binding domain mediated properties of the protein.

**Inhibition of SRSF1 oncogenic properties is through p38-MAPK activation.** Given that SRSF1 acts as an oncoprotein and its inhibition by decoy oligonucleotides activated the p38-MAPK stress pathway, we predicted that treatment of cancer cells with

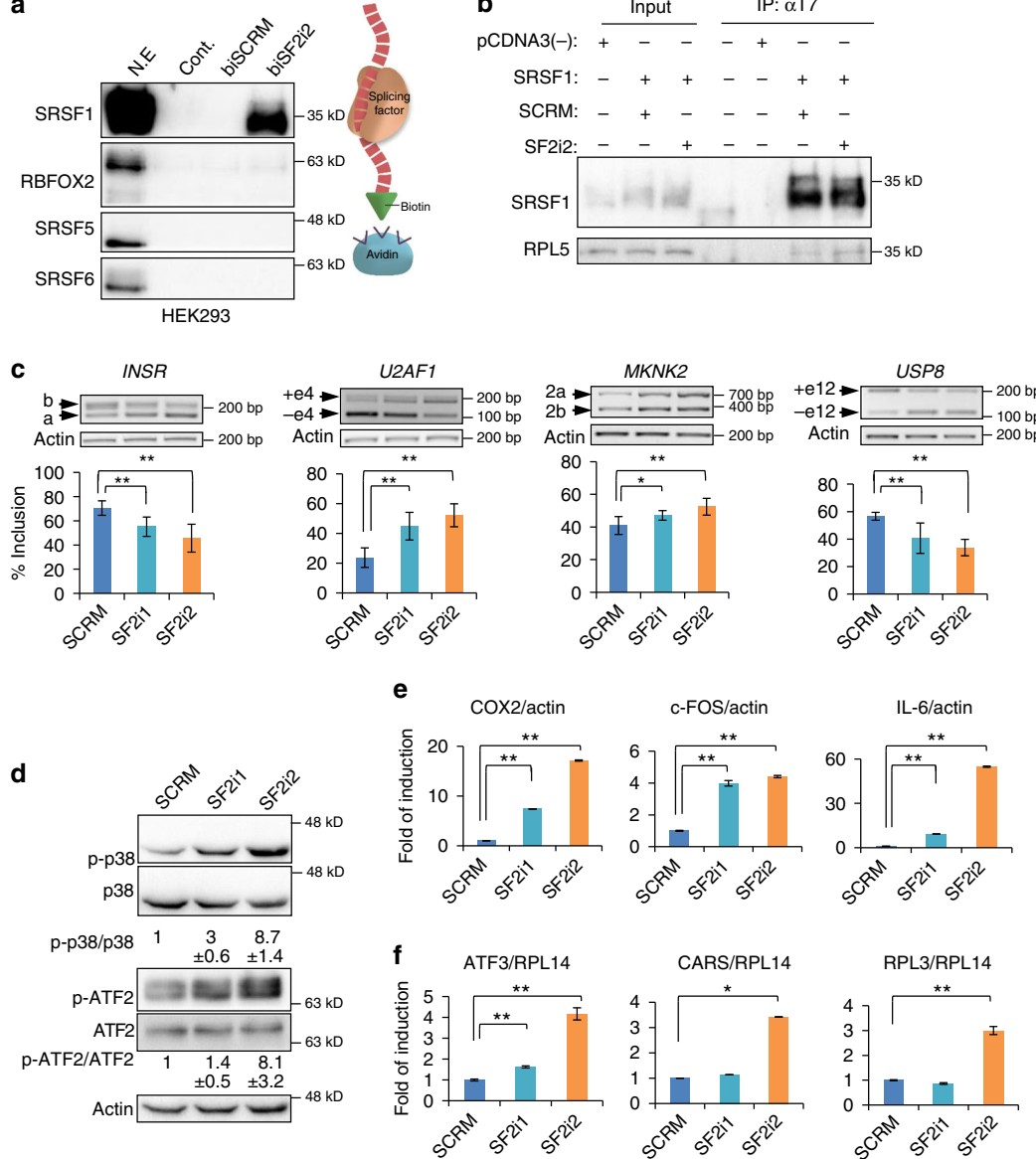

**Fig. 6** SRSF1 decoy oligonucleotides inhibit splicing and biological functions in glioblastoma cells. **a** Western blot of proteins pulled down with biotin conjugated SF2i2 (biSF2i2) or SCRM (biSCRM) using nuclear extracts from HEK293 cells overexpressing SRSF1. **b** Western blot of HEK293 cells co-transfected with pcDNA3 T7 SRSF1 and either SF2i2 or SCRM. Cells were fractionated and nuclear extract lysates were used for immunoprecipitation using anti-T7 antibody. Input and IPs were analysed by immunoblotting with the indicated antibodies. **c** RT-PCR and quantification of known splicing targets of SRSF1 in U87MG cells transfected with 2.5 μM of either SCRM, SF2i1, or SF2i2. Data represent means ± SD of six biological samples. **d** Western blot analysis of lysates from cells described in **c**. Data represent means ± SD of three biological samples. **e** RT-qPCR of p38 pathway target genes in cells described in **c**. Data represent means ± SD of triplicates. **f** RT-qPCR of NMD targets in cells described in **c**. *p-value ≤0.05, **p-value <0.004. Data represent means ± SD of triplicates. p-values were calculated using Student's t-test (two-tailed)

SRSF1 decoy oligonucleotides would reverse the transformed phenotype of these cells. Indeed, treatment of U87MG and MDA-MB-231 cells with SF2i1 and SF2i2 inhibited proliferation (3 to 4-fold), colony survival in a clonogenic assay and colony formation in soft agar while RBFOXi had no effect in these assays (Fig. 7a–c and Supplementary Fig. 10A-D). In order to confirm the specificity of this effect we performed the same experiments using three unrelated control oligonucleotides; Cont.1, Cont.2, Cont.3. As expected, the control oligonucleotides had no effect on the transformed phenotype of the cells, while SF2i2 had an obvious effect (Supplementary Fig. 11A-C). To verify that the observed reduced oncogenic activity is due to activation of the p38-MAPK

stress pathway, we used the p38-MAPK inhibitor SB203580 (Fig. 7d). Cells transfected with SF2i1 and SF2i2 formed more colonies in a soft agar assay and had increased survival in a clonogenic assay (4-fold) after treatment with SB203580 compared to cells treated with vehicle (Fig. 7e, f and Supplementary Fig. 11D-G). A concurrent reduction in ATF2 phosphorylation and decreased expression of downstream targets COX2, c-FOS, and IL-6 was also observed (Fig. 7g, h and Supplementary Fig. 11H). These results demonstrate that SRSF1 decoy oligonucleotides can inhibit malignant properties of glioblastoma and breast cancer cells through activation of the p38-MAPK stress pathway.

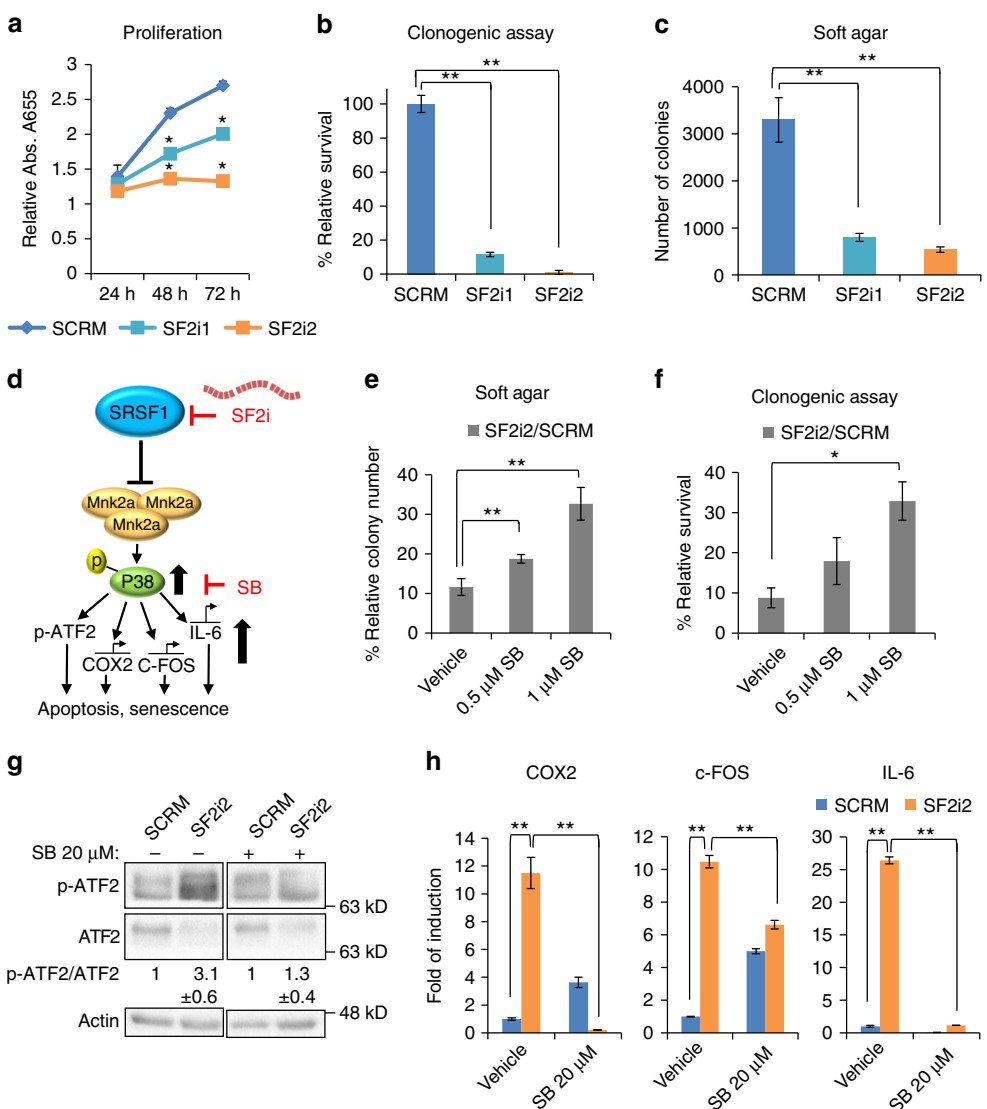

**Fig. 7** Inhibition of glioblastoma cells by SRSF1 decoys is partially reversed by SB203580. **a** Proliferation assay of U87MG cells after transfection with 2.5 μM of indicated decoy oligonucleotides. Four hours after transfection 4000 cells/well were seeded in six replicates. *p-value ≤0.02 for SF2i2 at 24 h, **p-value ≤0.0004 at 48 and 72 h. Data represent means ± SD of six replicates. **b** Graph of clonogenic assay (1000 cells/well) of transfected cells described in **a**. **p-value <0.003. Data represent means ± SD of four independent biological samples. **c** Quantification of soft agar colony growth assay on transfected cells described in **a**. Graphs represent quantification of 10 fields counted in duplicate (total of 20 fields). **p-value <1.1E–17. Data represent means ± SD of two wells. **d** Scheme representing SRSF1 effects on the p38-MAPK signaling pathway. SF2i1/SF2i2 oligonucleotides (SF2i), SB203580 (SB). **e** Quantification of soft agar colony growth assay on transfected cells described in **a** treated with the indicated concentrations of SB. Relative survival is based on quantification of 10 fields counted in duplicate (total of 20 fields). **p-value <0.002. Data represent means ± SD of two wells. **f** Quantification of clonogenic assay of cells described in **a** treated with the indicated concentrations of SB. *p-value = 0.02. Data represent means ± SD of duplicates. **g** Western blot analysis of cells described in **e**. Quantification of SF2i2 + vehicle is normalized to SCRM + vehicle, and SF2i2 + SB is normalized to SCRM + SB. Data represent means ± SD of four biological samples. **h** RT- qPCR of p38 pathway targets in cells described in **e**. Values are normalized to actin and SCRM without vehicle is arbitrarily set at 1. *p-value = 0.02, **p-value ≤0.003. Data represent means ± SD of triplicates. p-values were calculated using Student's t-test (two-tailed)

**SRSF1 decoys inhibit glioblastoma tumor growth in mice**. We next wanted to test if SRSF1 decoy oligonucleotides can inhibit glioblastoma growth in vivo. U87MG cells were labeled with mCherry and transfected with SF2i2 decoy oligonucleotides. Twenty-four hours after transfection cells were injected into the murine striatum of both brain hemispheres. Three weeks after the injection the mice were sacrificed and the tumors visualized under a fluorescent stereoscope. The mice injected with cells transfected with the SF2i2 decoy oligonucleotides developed notably smaller tumors than the control group transfected with

the SCRM oligonucleotides (Fig. 8a–c). These results suggest that decoy oligonucleotides against SRSF1 can inhibit glioblastoma tumor growth in vivo.

## Discussion

In this report we demonstrate that decoy oligonucleotides designed against specific splicing factors can bind the splicing factor, affect alternative splicing of known targets and have biological effects in line with inhibition of their splicing activity.

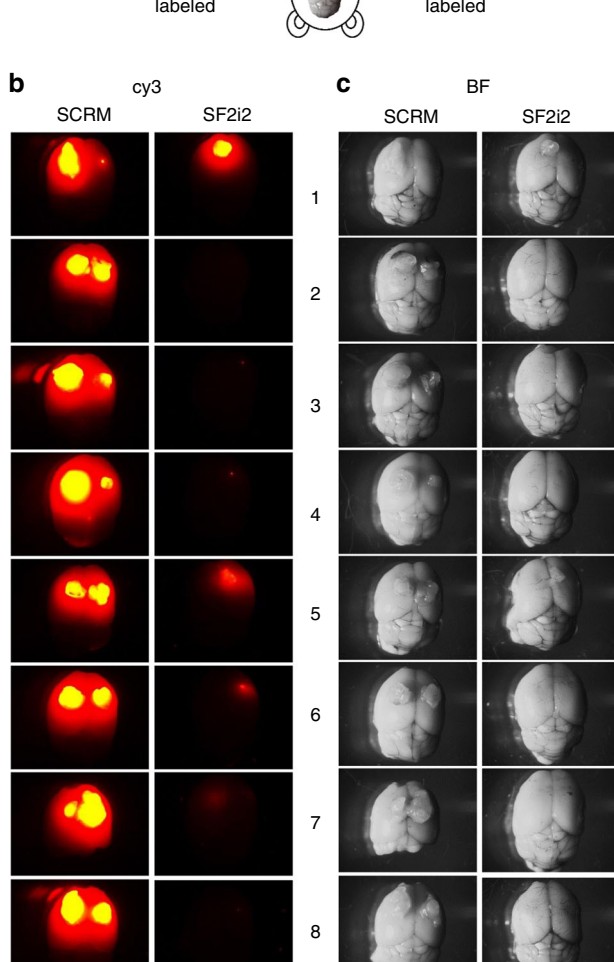

**Fig. 8** SRSF1 decoy oligonucleotides inhibit glioblastoma tumor growth in mice. **a** mCherry-labeled U87MG cells transfected with either SCRM or SF2i2 decoy oligos were injected intracranially into the striatum on both sides of the brain. Twenty-one days after injection the mice were sacrificed and brains were photographed using a fluorescent dissecting microscope. **b** Cy3 imaging. Exposure time was 500 ms. **c** Bright field photos (BF)

There are multiple advantages of the modified RNA decoy oligonucleotides presented here over existing technologies, such as siRNA or antisense oligonucleotides. One of the advantages is that the decoy oligonucleotides target the RNA binding domain of the splicing factor, inhibiting only its RNA binding activities without interfering with its other activities, such as protein–protein interactions. This is in contrast to knockdown technologies, which inhibit the expression of the protein and thereby inhibit all the activities of the splicing factor (some of which can be RNA-independent). Thus, it is expected that decoy oligonucleotides will have fewer side effects than complete knockdown, as suggested by the smaller amount of splicing events affected by the decoy against RBFOX1/2 than RBFOX1/2 knockdown (Fig. 2b). Another advantage is that these oligonucleotides are not dependent on the half-life of the splicing factor (and thus can act immediately as seen in Fig. 2a) or on any cellular mechanism that can be mutated in certain cancer cells (e.g., siRNAs, which depend on the cellular machinery).

Moreover, as some splicing factors are highly conserved through evolution, the decoy oligonucleotides might inhibit splicing factors in organisms, other than mammals, where there is no RNAi machinery present (e.g., plasmodium falciparum)[50]. Finally, the decoy oligonucleotides are single stranded RNA molecules, which have been shown to be successfully delivered into organs of the body[51] much more efficiently than double stranded siRNA molecules[6].

In our attempt to understand the biochemical basis for decoy oligonucleotide binding to the RRM of a specific splicing factor, we determined that the affinity of a decoy oligonucleotide with four repeats of the binding motif is almost five times greater than an oligonucleotide with a single binding motif. The explanation for this increased affinity probably resides in the fact that the additional motifs within the decoy oligonucleotide increased the number of splicing factor molecules bound to the oligonucleotide, suggesting cooperative binding of these proteins (Fig. 3). It has previously been shown that mutant splicing factors, for which the mutation decreased their affinity to the target RNA motif in vitro by a factor of five, had reduced splicing activity in cells[42,52].

In this paper, we demonstrate the successful use of decoy oligonucleotides targeting three different splicing factors; RBFOX1/2, PTBP1, and SRSF1. The in vivo effect of the RBFOX1/2 decoy oligonucleotides on splicing patterns and muscle development in zebrafish were very significant. The changes observed after oligonucleotide introduction into cells did not interfere with protein–protein interactions between RBFOX2 and hnRNP M, emphasizing the preciseness of this method. Similar to the decoy oligonucleotides against RBFOX1/2, we did not observe any interference with SRSF1 protein–protein interactions, such as the SRSF1-RPL5 complex, while other RRM related functions, such as NMD, were inhibited. Taken together, these results show that the mode of action of the decoy oligonucleotides is specific and affects RNA binding-mediated functions of the splicing factor. In contrast to the RBFOX1/2 decoy oligonucleotides, which showed specific binding to RBFOX1/2 proteins in the proteomic analysis, the SRSF1 decoy oligonucleotides bound multiple proteins in addition to SRSF1. This lack of specificity could be attributed to the SR protein family which is known to have redundant activity for some substrates in vitro and in vivo[53,54]. Another explanation is the limitation of the proteomic assay. This assay cannot distinguish between proteins that bind directly to the decoy oligonucleotides and indirect binders that bind SRSF1 through protein–protein interactions. A large number of proteins identified in the SRSF1 proteomic analysis have been recently demonstrated to bind SRSF1 through protein–protein interactions[40]. Even though the binding motif of some splicing factors is not well defined, we observed high specificity to specific splicing targets. Several SRSF1 targets were solely affected by the SRSF1 decoy oligonucleotides and not by the PTBP1 and RBFOX1/2 decoy oligonucleotides and vice versa, the SRSF1 decoy oligonucleotides did not affect splicing of some of the PTBP1 and RBFOX1/2 splicing targets tested (Supplementary Figs. 8 and 9). Moreover, only the SRSF1 decoy activated p38-MAPK while all the other decoy and control oligonucleotides did not (Supplementary Fig. 9). The decoy oligonucleotide against PTBP1 was successful in inhibiting the oncogenic properties of the cells, which opens up the possibility of treatment of cancer with these types of decoy oligonucleotides, targeting splicing factors that are upregulated in certain cancer types. We also show that inhibition of SRSF1 with decoy oligonucleotides has an effect on the oncogenic properties of cancer cells and identified that this occurs through activation of the p38-MAPK pathway. Finally, glioblastoma cells transfected with SF2i2 and injected into the mouse striatum developed smaller tumors, or no tumors, compared to

SCRM transfected cells. These results demonstrate the clinical potential of the decoy oligonucleotide strategy.

Inhibition of pleotropic splicing factors such as PTBP1, SRSF1, or RBFOX1/2 is expected to affect the splicing of hundreds of genes[25]. Thus, the question of toxicity of their inhibitors is a major concern for such an approach. However, many oncogenes and oncogenic pathways (e.g., c-Myc, EGFR, Akt, mTORC1, Ras/Raf, and others), which regulate normal growth, are valid targets for cancer therapy. The question of toxicity due to inhibition of each of these splicing factors and others must be evaluated carefully. Another complexity in inhibiting splicing factors is that they function in many steps of RNA processing; splicing, mRNA export, NMD, and translation[55]. Thus, even a specific inhibition of their RNA binding activity can affect several different functions. Some of these functions may be beneficial for inhibition of their oncogenic activity and others may not. The advantage of competitive decoy oligonucleotides over knockdown methods is that the level of inhibition can be better controlled in a dose-dependent manner. In addition, the decoy oligonucleotides do not interfere with other functions of the splicing factors (not related to RNA processing), a fact that is expected to reduce the toxicity of this approach compared to knockdown of the splicing factor.

In conclusion, the decoy oligonucleotide technology presented here is the first type of intelligence-based design of molecules that can directly and specifically inhibit the activity of RNA binding proteins (RBPs), such as splicing factors; a substantial advantage over traditional knockdown methods.

## Methods

**Confocal imaging**. Fixed cells images were taken using a Zeiss LSM710 confocal microscope and processed using ZEN lite 2011 blue edition (v2.3). Live cells images were taken by Nikon SMZ18 stereomicroscope using NIS–Elements Br software.

**Cells**. U87MG, MDA-MB-435S, MDA-MB-231, and HEK293 cells were grown in DMEM supplemented with 10% (v/v) FCS, penicillin, and streptomycin.

**Immunoblotting**. Cells were lysed in Laemmli buffer and analyzed for total protein concentration[56]. Fifteen microgram of total protein from each cell lysate was separated by SDS-PAGE and transferred onto a PVDF membrane (Invitrogen). The membranes were probed with primary antibodies. Primary antibodies: RBFOX2 (1:4000, Sigma), SRSF1 (1:200, mAb AK96 culture supernatant[57]), SRSF5 (1:3000, Sigma), SRSF6 (1:500, mAb 8-1-28 culture supernatant), OctA (Flag tag) (1:500, Santa Cruz), hnRNPM (1:500, Novusbio), PTBP1 (1:10,000, Abcam), T7 (1:5000 Novagen), phospho-p38 (Thr180/Tyr182) (1:1000 cell signaling), p38 (1:1000 Santa Cruz), phospho- ATF2 (Thr71) (1:1000 Cell Signaling), ATF2 (1:1000 abcam), β tubulin I + II (1:1000, Sigma), β-actin (1:200, Santa Cruz), GAPDH (1:1000, Santa Cruz), Secondary antibodies: HRP-conjugated goat anti-mouse, goat anti-rabbit, or donkey anti-goat IgG (H + L) (1:10,000, Jackson Laboratories).

**Immunoprecipitation**. HEK293 cells were transfected with pCDNA3-Flag-RBFOX2 (received from the Krainer Lab, Cold Spring Harbor Laboratory), pCDNA3-T7- SRSF1[58], or pCDNA3(−). Twenty-four hours after plasmid transfection, cells were transfected with oligonucleotides (RBFOXi, SF2i2 or SCRM). Forty-eight hours after oligonucleotide transfection cells were fractionated and lysed with in nuclear extract buffer (described in pull down assay). One microgram of anti-Flag or anti-T7 antibody bound to 50 μl of 50% protein G-sepharose slurry was incubated with 500 mg of nuclear extract overnight. After washing four times with nuclear extract buffer, beads were incubated with 50 μl of 2× Laemmli buffer and separated by SDS-PAGE.

**RT-PCR**. Total RNA was extracted with TRI Reagent (Sigma) and 1 μg of total RNA was reverse transcribed using High Capacity cDNA Reverse Transcription Kit (Applied Biosystems). PCR was conducted on 1 μl of cDNA by KAPA 2G Fast HS ReadyMix PCR kit (KAPA Biosystems). PCR conditions were as described in manufacturer's protocol of ReadyMix with the addition of 5% (v/v) DMSO for either 30, 35, or 40 cycles. PCR products were separated on 2% agarose gels. Primers listed in Supplementary Table 1.

**RT-qPCR**. Total RNA was extracted with TRI Reagent (Sigma) and 1 μg of total RNA was reverse transcribed using High Capacity cDNA Reverse Transcription Kit (Applied Biosystems). qPCR was performed on the cDNA using SYBR green (Applied Biosystems) and the CFX96 (Bio-Rad) real-time PCR machine. Normalization was performed using either β-actin or RPL14 and SCRM value was arbitrarily set to 1. Samples were compared to a standard curve, which was established by serial dilutions of a known concentration of cDNA. The PCR reaction is composed of the following steps: one cycle for 30 s at 95 °C and 39 cycles of 5 s at 95 °C and 30 s at 59–63 °C according to Tm of primers. Primers are listed in Supplementary Table 1.

**Transfection of oligonucleotides**. Cells were transfected with oligonucleotides using lipofectamine 2000 (Invitrogen) according to manufacturer's protocol.

**Stable cell lines**. To generate stable cell lines, U87MG and MDA-MB-231 cells were transduced with MSCV-puro- shRNA retroviral vectors against RBFOX2 (received from the Krainer Lab, Cold Spring Harbor Laboratory). Medium was replaced 24 h after infection, and 24 h later, infected cells were selected with puromycin (2 μg/ml) for 72 h.

**Anchorage-independent growth**. Twenty-four hours post-transfection 15,000 cells per well were seeded in duplicates in 6-well plates. Each well was coated with 2 ml of bottom agar mixture (media, 10% FBS, 1% agar). After the bottom layer had solidified, 2 ml of top agar mixture (media, 10% FBS, 0.3% agar) containing the cells was added. After this layer had solidified, 2 ml of media (media, 10% FBS) was added. Plates were incubated at 37 °C with 5% carbon dioxide. Following 10–21 days, colonies from ten different fields were counted and the average number of colonies per well was calculated.

**Growth curves (proliferation)**. Four thousand cells per well were seeded in 96-well plates 4 h post-transfection in six replicates, fixed and stained with methylene blue. Cell density was determined every 24 h (until 72 h/96 h) by absorbance of methylene blue staining at 655 nm measured on a plate reader (BioRad).

**Clonogenic assay**. 1000, 500, and 250 cells were seeded in 6-well plates. After 10–21 days, cells were fixated with 2.5% glutaraldehyde solution for 10–15 min and stained with 1% methylene blue solution.

**SB203580 treatments**. In experiments using the MAPK pathway inhibitor, SB203580, the reagent was added 4 h after transfection and remained until the end of the assay. For anchorage independent growth assays and clonogenic assays the concentrations were 0.5 and 1 μM, for Western blot assays and RT-qPCR the concentration was 20 μM.

**Pull down assay**. Nuclear extracts (NE) were lysed in a buffer containing 20 mM Hepes, 1.5 mM McCl₂, 420 mM NaCl, 0.2 mM EDTA and 25% (v/v) glycerol. NE was incubated with biotin tagged oligonucleotides. Streptavidin beads (Thermo Scientific), previously blocked with a solution containing heparin, were added after 3 h. The pulled down fraction was lysed in Laemmeli buffer and analyzed by western blot analysis.

**Zebrafish injections**. Tupfel Long-fin (aka TL) zebrafish breeding and maintenance were performed as previously described[59]. 5 or 8 pg of oligonucleotides were injected into zebrafish embryos at the 1–2-cell stage. Injected embryos were allowed to develop at 28.5 °C. Forty-eight hours post fertilization embryos were fixed in 4% paraformaldehyde for immunohistochemistry or solubilized in TRI Reagent (Sigma) and analyzed as described for RT-PCR. All experiments complied with ethical regulations for animal testing and research of the Weizmann Institute and IACUC.

**Intracranial injections**. U87MG-mCherry cells were injected via stereotactic surgery into the murine striatum in both hemispheres, 2 mm lateral and to a depth of 3 mm to the bregma. Each injection consisted of 2 μl of solution, containing 200,000 cells/μl transfected with decoy oligonucleotides. The mice were sacrificed after three weeks, the brains were extracted and visualized using Nikon SMZ18 stereomicroscope and NIS–Elements Br software.

All animal experiments were performed in accordance with the guidelines of the Hebrew University committee for the use of animals for research. All experiments complied with ethical regulations for animal testing and research, Hebrew University (IACUC) Ethics approval no' MD-15-14634-5. HU is AAALAC approved.

**Immunohistochemistry**. Embryos were fixed overnight in 4% paraformaldehyde and permeabilized with triton. Alexa Fluor 594 conjugated phalloidin (Invitrogen) was used at 1 unit/ml following manufacturer's instructions. Embryos were mounted in 80% glycerol and visualized using fluorescent binocular microscope.

**RNA-seq**. Total of 1 μg RNA was used to prepare libraries using Illumina library kits. $100 \times 10^6$ reads of 100 bp from each side (paired-end) were generated for every samples using the Illumina HiSeq machine. STAR aligner (version 2.4.2a) (Dobin et al. 2013) was used to align each read uniquely to the Hg19 human genome (default parameters except alignSJoverhangMin 8, alignSJDBoverhangMin 3, outFilterMismatchNmax 999, alignIntronMin 20, alignIntronMax 1000000, alignMatesGapMax 1000000, outFilterMismatchNoverReadLmax 0.1). To determine differentially spliced exons between the samples we used rMATs (version 3.0.9) using human UCSC known genes annotation. The statistical analysis and comparison with Damianov et al. results were done using R (the R Project for Statistical Computing (http://www.r-project.org/)).

The RNA-seq was performed on two independent biological replicates of cells transfected with SCRM or RBFOXi.

**Proteomic analysis**. RBFOXi, PTBP1i, and SF2i interacting proteins were subjected to on-bead digestion. The complexes were reduced with 1 mM Dithiothreitol in 2 mM urea for 30 min, followed by alkylation with 5 mM Iodoacetamide in 2 mM urea for 30 min in the dark. Proteins were digested overnight with sequencing-grade trypsin (Promega) and digestion was terminated by adding 0.1% trifluoroacetic acid (TFA). The peptides were desalted and concentrated on C18 stage tips[60]. Prior to MS analysis, peptides were eluted from the stage tips using 80% acetonitrile, vacuum-concentrated and diluted in loading buffer (2% acetonitrile and 0.1% trifluoroacetic acid) and subjected for MS measurements. LC-MS/MS analysis was performed using nano-ultra high performance liquid chromatography (nano-UPLC; Easy-nLC1000; Thermo Scientific) coupled on-line to a Q-Exactive Plus mass spectrometer (Thermo Scientific). Peptides were eluted using 240-min gradient of water:acetonitrile. Raw files were analyzed with Max-Quant (1.5.3.36) and the Andromeda search engine[61,62]. MS/MS spectra were searched with reference to human UNIPROT database and a decoy database to determine false discovery rate (FDR). FDR thresholds were set to 0.01 for protein and peptide identification. All the statistical analyses of the MaxQuant output tables were performed with the Perseus software[63]. For each analysis ProteinGroups file was filtered to include only proteins that were identified in at least two (out of three) replicates of the pull down. We then imputed missing values by replacing them with a constant value that matches the lowest intensity in the dataset (RBFOXi) or by creating a normal distribution with a downshift of 1.8 standard deviations and a width of 0.3 of the original (PTBPi). Different imputation strategies were selected due to the significantly higher purity of the RBFOXi samples. Significant interactors were examined by performing one-sided T-test for RBFOX1/2, PTBP1 against SCRM with permutation-based FDR cut-off of 0.05 and $S0 = 0.1$[64] and SRSF1 against SCRM with FDR cut-off of 0.1 and $S0 = 0.3$.

**RBFOX1 RRM purification**. ORF of RBFOX1 RRM (amino acids 109–208) was cloned in the pET28a expression vector. The protein was overexpressed at 37 °C in E. coli BL21 (DE3) codon plus cells in minimal M9 medium containing 1 g/l $^{15}NH_4Cl$ and 4 g/l glucose, purified by two successive nickel affinity chromatography (QIAGEN) steps using an N-terminal 6× His tag, dialyzed against T40 (10 mM Tris-HCl pH 7.4, 40 mM NaCl, 0.05% Tween20) or T100 (10 mM Tris-HCl pH 7.4, 100 mM NaCl, 0.05% Tween20) buffer and concentrated to 0.3 mM with a 10-kDa molecular mass cutoff Centricon device (Vivascience).

**NMR experiments**. RNA oligonucleotides were purchased from Dharmacon, deprotected according to the manufacturer's instructions, lyophilized and resuspended in T40 buffer. The NMR titrations were all performed in the NMR buffer at 313°K on a Bruker AVIII-600 MHz spectrometer equipped with a cryoprobe. Data were processed using Topspin 2.1 (Bruker) and analysed with Sparky (http://www.cgl.ucsf.edu/home/sparky/).

**SwitchSENSE experiments**. For this study, we used long DNA/RNA chimeric sequences (around 60–80nts) for which the 3′-extremity is a ssDNA molecule fully complementary to the 48-nts long ssDNAs attached to the electrodes and the 5′-part corresponds to a flanking non-hybridized sequence containing a linker of 4 thymines followed by the RNA motif bound by the protein (Fig. 3a). DNA/RNA chimeric molecules were purchased from ELLA Biotech GmbH, resuspended in T40 buffer at 500 nM and hybridized to the biochip as illustrated in Fig. 3a. RBFOXi ×1 to ×4 molecules contained one to four consecutive RBFOX1 binding sites and SCRM ×1 to ×4 contained random sequences of similar lengths (Supplementary Table 1). All switchSENSE experiments were performed on a DRX analyser using MPC-48-2-Y1-S chips (both Dynamic Biosensors GmbH, Martinsried, Germany). Sizing experiments were performed in T40 buffer with an association time of 20 min at a flow rate of 50 μl/min and a RBFOX1 RRM concentration of 150 nM to saturate with proteins the RNA molecules attached to the biochip. We verified that at this protein concentration no interaction was observed with scrambled equivalent RNAs (SCRM ×1 to ×4). For sizing, the time-resolved fluorescence intensity signal corresponding to the upward motion of the DNA nanolevers was evaluated as described by Langer et al.[65,66] using switchANALYSIS software (Dynamic Biosensors GmbH, Planegg, Germany). The size analysis

allowed estimation of an apparent size value for protein-RNA complexes. Titration experiments were performed in T100 buffer using an association time of 20 min at a flow rate of 10 μl/min and increasing RBFOX1 RRM concentrations (0.3, 0.6, 1.2, 2.3, 4.7, 9.4, 18.8, 37.5, 75, 150 nM). Normalized fluorescence intensity signals were plotted against protein concentrations and $K_D$ values were from the curves by a global fit using a 1:1 Langmuir fit model:

1:1 Langmuir Equation:

$$\mathrm{fb}_{eq}(c) = \mathrm{fb}_0 + (\mathrm{fb}_1 - \mathrm{fb}_0) * \frac{c}{K_D + c},$$

$\mathrm{fb}_{eq}$ is the fraction bound in equilibrium; $\mathrm{fb}_0$ is the fraction bound at lowest concentration; $\mathrm{fb}_1$ is the fraction bound at highest concentration; $c$ is the concentration; $K_D$ is the equilibrium dissociation constant.

All measurements were performed at 37 °C.

Unprocessed data: Unprocessed images of RT-PCR agarose gels and Western blots appear in Supplementary Fig. 12.

**Reporting summary**. Further information on experimental design is available in the Nature Research Reporting Summary linked to this article.

## Data availability
RNA-seq data have been deposited to the GEO repository as GEO submission GSE126503. The mass spectrometry proteomics data have been deposited to the ProteomeXchange Consortium via the PRIDE [1] partner repository with the dataset identifier PXD012564. The authors declare that all the other data supporting the findings of this study are available within the article and its supplementary information files and from the corresponding author upon reasonable request. The source data underlying Figs. 1b, 1d, 4b, 5a, 6a, 6b, and 6d, are provided in Supplementary Fig. 12.

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

## Acknowledgements

The authors wish to thank Dr. Zahava Kluger for comments on the manuscript and members of the Karni lab for helpful discussions. This work was supported by the Israel Science Foundation (ISF Grants no' 1290/12 to R.K.), Alex U. Soyka Pancreatic Cancer Research Project (to R.K.) and KAMIN (Israel Innovation Authority) grant (to R.K.). We thank the SNF-NCCR RNA and Disease for financial support to F.H.T.A and A.C.

## Author contributions

P.D. and R.K. designed the experiments. P.D., M.M., J.B., G.D.B., A.C., and S.K. performed experiments. O.S., E.Y., J.B., G.L., F.H.A., T.W., and T.G. contributed reagents and technical help. M.D.-G. and E.Y.L. analyzed RNAseq data. P.D. and R.K. analyzed the data and wrote the manuscript.

## Additional information

**Competing interests:** The authors declare no competing interests.

