## [Peer Review File · Nature Communications]

Reviewers' comments:

Reviewer #1 (Remarks to the Author):

In their manuscript "Specific Inhibition Of Splicing Factor Activity By Decoy RNA Oligonucleotides", Denichenko and colleagues describe a novel and potentially very exciting development in the use of antisense technology, which could open a new therapeutic avenue in the treatment of diseases where RNA-binding protein play a causative and/or important role. For example, many splicing factors show strong oncogenic properties and are overexpressed in tumors, and the role of some RBP in neurodegenerative diseases is well established.

While oligonucleotides have been used as decoys to affect the biological activities of transcription factors (e.g. STAT3 and others), miRNAs, siRNAs and some RNA-binding proteins (REV), their application to the functional knock-down of splicing factors' activities is novel and of great potential impact.

The authors identify three different splicing factors for which high-affinity binding motifs have been described: RBFOX1/2, PTBP1 and SRSF1. They then use modified RNA ASOs encompassing multiple copies of these consensus motifs to show that the ASOs are recognized by their cognate RBPs with some degree of specificity, and that such decoy treatment can affect significantly the molecular and biological activity of the target SF in vitro (all three) and also in vivo (SRSF1) in an orthotopic model of GBM.

The biological effects in vitro and especially in vivo are impressive, and could open the way to a new class of antisense-based drugs.

An important aspect of viewing these compounds as therapeutic leads (especially as representative of a class of compounds) is their specificity and the potential for toxicity due to both off-target and on-target effects (since the factors are important also in normal cells). The authors address the specificity by performing pull-down experiments using biotinylated variants of the compounds, followed by mass-spec analysis. In the case of RBFOXi, RNAseq experiments were also performed to look at the global splicing effects.

Unfortunately, the ability to evaluate these important experiments was undercut by the fact that the files provided were lacking a large part of the data set (supplementary table S1, and to a lesser extent Supplementary Table S2), probably because of some mishap occurred during submission of the manuscript.

More specifically, Supplementary Table S1 is not clearly designated as such in the file that I received, nor there is any detailed description of its contents, although it is reasonable to assume that it is the large table expanding from page 41 to 64 of the manuscript. Other than lacking a description, the table is not easily legible because each row is spread across various non-contiguous pages, without any obvious reference points to put it back together, and –more importantly- there is no indication about what the values reported in the table indicate (that is, there are no column headers). The crucial point, however, is that table S1 should contain the data on the mass-spec pull-down experiments with RBFOXi (page 5), PTBP1i (page 7) and SF2i2 (page 8), but it seems that the majority of the table is missing and only some (partial) data on only one of the experiment is shown (likely SF2i2, based on some of the values).

Clearly, it is impossible to properly evaluate the specificity claims, nor any conclusions the author's draw from the mass-spec data, since the relevant data-set data cannot be examined by the reviewer (or the reader).

Table S1 should be re-uploaded in its complete form, include a clear description of the various fields/values, and reformatted so that it will be more legible. I suggest to attach it as a separate PDF of the original spreadsheet rather than a broken-up table in the text file. Even better, three separate tables with the data from the three experiments (RBFOXi, PTBP1i, SF2i2) should be provided, rather than one giant table.

I would also suggest that in the revised manuscript a 'digest' of the most relevant data from the mass-spec experiments is included as a small table in the main body of the article.

Table S2 suffers, to a lesser extent, from some of the same issues. In particular, also in this case there is no clear designation nor any description of the table, and while the text claims that Table S2 reports all 73 splicing events affected by RBFOX1, only 49 are actually included in my copy (presumably the intersection with the events following RBFOX knockdown, as described by Danilov and colleagues, but this is never expressly stated). In any case, all 73 events should be reported, the 49 that intersect the other dataset highlighted, and the meaning of the various column values explained (while some are self-explanatory, others are not).

In addition to the issue with the supplementary table, there are a few other issues that should be addressed in the revised manuscript.

- Figures 1c, 4b, and Suppl3a. These figures represent the results from the mass-spec analysis and while I am sure they will make more sense when the complete tables are provided, they still should be better explained, both in the text and in the figure legend (and the procedure used to generate them should be better described in the Methods section). Normally in a volcano plot the X-axis reports magnitude changes (typically fold change), and the Y-axis relates to significance of change. It is unclear what the X-axis is representing in this case, as t-test usually also refers to significance rather than magnitude. All three 'volcano' plots reported are extremely asymmetric. Although this is not impossible, it is also not the common appearance of such graphs. Does it mean that there are no proteins at all which bind better to the control SCRM compound than to any of the three test oligos? This seems unlikely. Also, it is unclear to me what criteria were used to determine the "cutoff significance curve" that isolates RBFOX1/2 (and the equivalent curves in the two other plots). As is, it seems rather arbitrary and with different parameters from the curves used later in the analogous plots (Fig 4b and Fig S3a). Without the benefit of looking at the data-sets, at least three of the points in Fig 1c appear to have a p -value < 0.01 and a change comparable or even better than that of RBFOX. What are they and why are they not significant? In the case of PTBP1 and SRSF1 there are in fact dozens of proteins that are above the indicated curve, and for SRSF1 the vast majority has a position that would indicate a higher binding specificity for SFi2 than SRSF1 itself.

It would also be useful if some key proteins were identified in the figure, (e.g., hnRNPM, PTBP1, SRSF1 in all three, and others), as well as the other points with the higher significance/fold-change values (again, this could be partially solved by the presence of a proper table S1, but it would still be valuable to have the id somewhere in the main text, rather than guess them by looking at the coordinates in the table).

- Pull-down vs. co-IP experiments. Since the pull-down experiments aim at identifying direct binders of the oligos, while co-IPs look for the interactors of the RBPs (e.g. RBFOX2 vs. hnRNPM) the pull-down conditions should be more stringent, to eliminate the interactors from the result pool of the pull-down. From the description in the text/methods, they seem the same. Also, was the pull-down of hnRNPM tested in the experiment described in figure 1b? Could this be used as a control to calibrate conditions for the two sets of experiments? More stringent pull-down conditions would also likely help to significantly clean-up the pull-down data, especially for SRSF1.

- Cell-based assays. While overall the results of these experiments are exciting and convincing, their display and organization could use some improvement. The authors switch continuously between U87MG and MBA231 cell lines. Most of the experiments are done with both cell lines with mostly comparable results, and one of the two is relegated to the supplementary section without a clear rationale. It would be helpful if there were more consistency in which data are used and where, or if a rationale were provided for doing so. In addition, while the cell lines used are clearly indicated in the figure legend, to also indicate them in the figures themselves would make the paper clearer to the reader.

- Clonogenic Assay. This assay is done on multiple cell lines and with various treatments, again the results are nice, but the data are displayed in a very inconsistent and confusing way. The assay itself is referred to either as "colony assay" (4e, s4d) or "colony survival" (6b, 6f, S4b, S5b, S5d). The Y-axis sometimes represents the number of colonies (4e, s4b, s4e, s5b), sometimes the % relative survival (6b), sometime the % normalized to the ratio of the treatment. The figure legends are equally variable and confusing. One consistent way to display the same type of data should be selected, and any change from it should be explained and well described in the figure legend. The rescue SB experiments are particularly inconsistently displayed and result a bit confusing (fig 6f vs. s5b vs. s5d).

Another (minor) inconsistency is the relative positioning of the soft agar vs. colony survival assay panels, which are swapped frequently, including within the same figure (fig 6).

- Figure 7. There seem to be some inconsistency in the mCherry signal intensity of the SF212-treated tumors in panel 1 and 8 comparing the CY3 and BF/CY3 images

Other points:

- The sequences of the decoy oligos should be specified in the text of the article and/or in one of the main figures (maybe Fig 1a?). An alignment with the consensus motifs of the relative protein should be included (since some have degenerate/multiple binding motifs). As is, the sequences are only included in supplementary table 3 (with the RT-PCR primers), and the consensus are not presented.

- The authors suggest (page 6, lines 3-5) that the discrepancy in the results between knock-down and functional inhibition by decoys may be accounted for by the RNA-binding-independent function of RBFOX1/2. This is a very intriguing hypothesis, which could be simply tested (at least initially) by verifying the presence of RBFOX1/2 binding sites in proximity of the splicing events affected in the two data sets.

- Both in the introduction and in the discussion, the authors assert that because the decoys bind to the RNA-binding domain (true), they will only affect the RNA-binding/splicing activity of the splicing factor (likely but not proven), and thus they are more desirable than -for example- knock-down approaches, because they would have fewer side effects, thus presumably being more specific. However, it is unclear why the targeting of SFs should be in principle limited to their "RNA binding/splicing" activities, and/or why this should necessarily be an advantage, since some of their pathogenic functions could well involve their non-RNA-binding properties. In some cases the opposite might even be true. Furthermore, such different activities are frequently connected and it might be complex to decouple them. The decoy approach is different from knock down strategies, and it might prove to indeed lead to a superior therapeutic strategy for at least some of the targets, but it is not inherently 'better', based on the above argument, than knock-down. On the other hand, some of the other arguments are convincing, such as that decoy might be more effective because they are more drug-like and do not depend on half-life and expression, that they can target a protein across species and can be delivered better. The discussion should be amended accordingly.

As far as specificity, whereas it is true that targeting just the RNA-binding activity might lead to "fewer side effects" than knocking down the protein tout-court as far as the intended target, it is also clear that the decoy could have much broader sequence-dependent off-target side effects, due to the redundant and degenerate binding motifs of SFs. For example, SF2i2 can clearly bind dozen of targets with apparent higher affinity than SRSF1 itself (fig S3a), which is likely not true for a well-designed siRNA.

- To address specificity, some splicing events could be analyzed that are predicted NOT to be affected by the decoy, for example, RBFOX targets from Fig 1 could be analyzed in PTBP1i-treated and Sf2i2-treated cells, and vice-versa with PTBP1i targets from fig 4 and SF2i2 targets from fig

5.

- Previous uses of decoy ASO strategies should be mentioned in the introduction even if (or especially because) they differ from the strategy proposed here.

- Some space in the discussion should be dedicated to address the possible toxicity/on-target side effects of this approach, since splicing factors like SRSF1 have broad functions also in normal cells.

- Figure legends should be fleshed out.

- In general, the text should be revised to improve the flow, as some parts still read as a draft and it seems overall a bit disorganized, particularly the introduction. For example, page 2, lines 3-4 is a verbatim repeat of page 1 lines 2-3; on page 3, line 10, the LASR complex should be referenced (ref 17) when it is defined; numerous typos are present throughout the manuscript.

Reviewer #2 (Remarks to the Author):

Splicing factors bind to specific RNA sequences. It is logical, therefore, that modified RNAs might be able to block splicing factors and selectively control their activities.

Comments

1) The premise of the paper is that RNAs that block splicing factors might be therapeutically important. This premise is weak. Compounds that block splicing factors would affect the splicing of many genes and lead to extensive off-target effects. Given that synthetic oligonucleotide complementary to specific gene targets struggle in the clinic with off-target effects, the notion that an therapeutic RNA might disrupt multiple gene expression events is not plausible.

2) This thrust to show that an anti-SF RNA might be a therapeutic adversely affects the structure of the paper by causing it to focus on weak in vivo experiments rather than rigorously making a case for the basic science hypothesis that anti-SF RNAs are possible.

3) If well-supported, the basic science showing that anti-SF RNAs are functional in cells would be suitable for well regarded journal like Nature Communications.

4) This paper supplies a wide array of data that are consistent with their RNAs having the desired effects. However, much of the data is superficial and unpersuasive. In this reviewer's opinion, a much more tightly focused paper focusing on supporting the fundamental hypothesis would have been preferred.

5) There are no biochemical experiments exploring the binding potency and specificity of SFs with RNAs. The hypothesis that a much longer than native 2'-O-methyl RNA can bind specifically to an SF is a remarkable one and requires beginning with simple biochemical assays.

6) Western analysis of pull down experiments is of limited value. One would expect that, at a high enough concentration, an RNA binding protein will bind its cognate sequence.

7) Mass spectrometry is not a trivial experimental technique. It is prone to experimenter bias and cherry-picking. The experimental description of the mass spectral analysis is entirely deficient, making it impossible to judge the rigor of the experiments.

8) It is not clear why an oligonucleotide with three or four binding motifs would associate with a splicing factor more strongly than an oligonucleotide with just one motif. The splicing factor can

bind just one motif at a time. While I do not exclude that it is conceivable that the design concept might be viable, the lack of precedent is a good example of why a much more careful biochemical framework is essential. There is no recognition that adding binding sites to increase affinity will seem illogical to many and needs to be carefully defended.

9) For experiments that examine anti-growth phenotypes upon addition of oligonucleotides, more controls are necessary. Many oligonucleotides have off-target anti-proliferative effects. This is a problem for both Figures 6 and 7. For Figure 7, oligos are notorious for having anti-tumor effects that are unrelated to binding to their intended targets inside cells.

10) How many copies of the decoy enter cells, are released from endosomes, and enter the nucleus? How many copies of splicing factors are there? Is the decoy effect plausible from the standpoint of molecular stoichiometry?

11) Oligos with PS backbones tend to be nonspecifically bind proteins, increasing the requirement for rigorous controls and cautious interpretation of experiments.

12) This review should not be construed as a complete list of the problems with this paper. The entire approach needs to be more carefully thought out from start to finish.

Remarkable results demand remarkable proof. A short, repetitive oligonucleotide that binds selectively to a splicing factor and acts as a decoy would be a remarkable result. The authors have fallen into the trap of putting together a paper that touches on many experiments but does none of them convincingly. complex experiments are presented with a gross lack of clear supporting data. Controls are minimal. A paper that focuses on only validating the basic hypothesis and does so convincingly would advance the field and be a better candidate for publication.

A point-by point answer to the reviewer's questions and comments:

Reviewer #1 (Remarks to the Author):

"In their manuscript "Specific Inhibition of Splicing Factor Activity By Decoy RNA Oligonucleotides", Denichenko and colleagues describe a novel and potentially very exciting development in the use of antisense technology, which could open a new therapeutic avenue in the treatment of diseases where RNA-binding protein play a causative and/or important role. For example, many splicing factors show strong oncogenic properties and are overexpressed in tumors, and the role of some RBP in neurodegenerative diseases is well established. While oligonucleotides have been used as decoys to affect the biological activities of transcription factors (e.g. STAT3 and others), miRNAs, siRNAs and some RNA-binding proteins (REV), their application to the functional knock-down of splicing factors' activities is novel and of great potential impact." " The biological effects in vitro and especially in vivo are impressive, and could open the way to a new class of antisense-based drugs." **–We thank the reviewer for sharing our enthusiasm regarding the potential and novelty of this approach.**

"An important aspect of viewing these compounds as therapeutic leads (especially as representative of a class of compounds) is their specificity and the potential for toxicity due to both off-target and on-target effects (since the factors are important also in normal cells). The authors address the specificity by performing pull-down experiments using biotinylated variants of the compounds, followed by mass-spec analysis. In the case of RBFOXi, RNAseq experiments were also performed to look at the global splicing effects. Unfortunately, the ability to evaluate these important experiments was undercut by the fact that the files provided were lacking a large part of the data set (supplementary table S1, and to a lesser extent Supplementary Table S2), probably because of some mishap occurred during submission of the manuscript.

More specifically, Supplementary Table S1 is not clearly designated as such in the file that I received, nor there is any detailed description of its contents, although it is reasonable to assume that it is the large table expanding from page 41 to 64 of the manuscript. Other than lacking a description, the table is not easily legible because each row is spread across various non-contiguous pages, without any obvious reference points to put it back together, and –more importantly- there is no indication about what the values reported in the table indicate (that is, there are no column headers). The crucial point, however, is that table S1 should contain the data on the mass-spec pull-down experiments with RBFOXi (page 5), PTBP1i (page 7) and SF2i2 (page 8), but it seems that the majority of the table is missing and only some (partial) data on only one of the experiment is shown (likely SF2i2, based on some of the values).

Clearly, it is impossible to properly evaluate the specificity claims, nor any conclusions the author's draw from the mass-spec data, since the relevant data-set data cannot be examined by the reviewer (or the reader). Table S1 should be re-uploaded in its complete form, include a clear description of the various fields/values, and reformatted so that it will be more legible. I suggest to attach it as a separate PDF of the original spreadsheet rather than a broken-up table in the text file. Even better, three separate tables with the data from the three experiments (RBFOXi, PTBP1i, SF2i2) should be provided, rather than one giant table. I would also suggest that in the revised manuscript a 'digest' of the most relevant data from the mass-spec experiments is included as a small table in the main body of the article." **–We truly apologize for this mistake in uploading of all of these supplementary tables. We have now arranged them as the reviewer suggested,**

separately (now Supplementary Tables 2, 4, 6) and will upload them fully onto the website so hopefully there will be no problem for the reviewer to access it in full.

"Table S2 suffers, to a lesser extent, from some of the same issues. In particular, also in this case there is no clear designation nor any description of the table, and while the text claims that Table S2 reports all 73 splicing events affected by RBFOX_i, only 49 are actually included in my copy (presumably the intersection with the events following RBFOX knockdown, as described by Danilov and colleagues, but this is never expressly stated). In any case, all 73 events should be reported, the 49 that intersect the other dataset highlighted, and the meaning of the various column values explained (while some are self-explanatory, others are not)." **-Again, we deeply apologize. The new table (now Supplementary Table 3) has two sheets; the first shows the 73 events identified in our experiment and the second shows the 49 events which are in common with the Damianov-Black paper.**

"...- Figures 1c, 4b, and Suppl3a. These figure represent the results from the mass-spec analysis and while I am sure they will make more sense when the complete tables are provided, they still should be better explained, both in the text and in the figure legend (and the procedure used to generate them should be better described in the Methods section). Normally in a volcano plot the X-axis reports magnitude changes (typically fold change), and the Y-axis relates to significance of change. It is unclear what the X-axis is representing in this case, as t-test usually also refers to significance rather than magnitude. All three 'volcano' plots reported are extremely asymmetric. Although this is not impossible, it is also not the common appearance of such graphs. Does it mean that there are no proteins at all which bind better to the control SCRM compound than to any of the three test oligos? This seems unlikely.

Also, it is unclear to me what criteria were used to determine the "cutoff significance curve" that isolates RBFOX_{1/2} (and the equivalent curves in the two other plots). As is, it seems rather arbitrary and with different parameters from the curves used later in the analogous plots (Fig 4b and Fig S3a). Without the benefit of looking at the data-sets, at least three of the points in Fig 1c appear to have a p-value < 0.01 and a change comparable or even better than that of RBFOX. What are they and why are they not significant? In the case of PTBP1 and SRSF1 there are in fact dozens of proteins that are above the indicated curve, and for SRSF1 the vast majority has position that would indicated a higher binding specificity for SFi2 than SRSF1 itself. **-We apologize for the misunderstanding. The volcano plots represent a one-sided T-test, and therefore are seen as extremely asymmetric. In addition, we use the S0 parameter [Tusher et al, PNAS 2001 pmid: 11309499], which balances between the T-test difference and the p-value. When using a value greater than 0, this parameter gives higher significance to the T-test difference than the p-value. Therefore, there may be cases with higher -log-p value that will not be significant due to lower T-test difference. Adding S0 parameter is very common as it increases the robustness of the results, which focus on the highly changing proteins. Importantly, the reviewer should note that we are using permutation-based FDR for multiple hypothesis testing correction. Therefore, the p-value in the graph does not directly correspond to the q-value that was used in each test. The q-values are now included in the Supplementary Tables.**

"It would also be useful if some key proteins were identified in the figure, (e.g., hnRNPM, PTBP1, SRSF1 in all three, and others), as well as the other points with the higher significance/fold-change values (again, this could be partially solved by the presence of a proper table S1, but it would still be valuable to have the id somewhere in the main text, rather than guess them by looking at the coordinates in the table)." **-We have now modified the volcano plots according to the reviewer's request and indicate the names of key proteins.**

"- Pull-down vs. co-IP experiments. Since the pull-down experiments aim at identifying direct binders of the oligos, while co-IPs look for the interactors of the RBPs (e.g. RBFOX2 vs. hnRNPM) the pull-down conditions should be more stringent, to eliminate the interactors from the result pool of the pull-down. From the description in the text/methods, they seem the same." **–The reviewer is correct. We perform these experiments in similar conditions, even though we probably should have try to add some additional detergents to the pull-down experiments to reduce the background. We tried to increase the salt concentration but had technical problems so we didn't continue to test additional salt concentrations. This probably increased the background of proteins that did not bind directly to the decoy oligonucleotides but rather were pulled-down by protein-protein interactions with the bound splicing factor. We refer to this now in several places in the text (see below), and we are aware that the actual specificity of the decoy oligonucleotides tested might be higher (as seen in the new Supplementary Fig. 8).**

"Also, was the pull-down of hnRNPM tested in the experiment described in figure 1b? Could this be used as a control to calibrate conditions for the two sets of experiments? More stringent pull-down conditions would also likely help to significantly clean-up the pull-down data, especially for SRSF1. **–We completely agree with the reviewer and we have tried to repeat all three proteomic pull-downs with higher salt concentrations (300mM) but had technical problems. We only succeeded with the PTBP1 pull down at this concentration (data not included in the new version). Thus, we remained with the experiments we had in the previous version. We are aware that there might be conditions where higher specificity can be shown. We now note in the text that many of the proteins pulled-down by SRSF1 were previously shown to bind to it in an RNA-independent manner and therefore we cannot distinguish between direct binding to the oligonucleotide and indirect binding (through protein-protein interactions) in the pull-down assays. We now explain this more directly in the text (see also below).**

"- Cell-based assays. While overall the results of these experiments are exciting and convincing, their display and organization could use some improvement. The authors switch continuously between U87MG and MBA231 cell lines. Most of the experiments are done with both cell lines with mostly comparable results, and one of the two is relegated to the supplementary section without a clear rationale. It would be helpful if there were more consistency in which data are used and where, or if a rationale were provided for doing so. In addition, while the cell lines used are clearly indicated in the figure legend, to also indicate them in the figures themselves would make the paper clearer to the reader." **–We agree and have now indicated the cell type (which is not the focus of the figure) in any figure which includes more than one cell type (e.g. blots of HEK293 cells in a figure containing splicing assays of U87MG cells).**

"- Clonogenic Assay. This assay is done on multiple cell lines and with various treatments, again the results are nice, but the data are displayed in a very inconsistent and confusing way. The assay itself is referred to either as "colony assay" (4e, s4d) or "colony survival" (6b, 6f, S4b, S5b, S5d). The Y-axis sometimes represents the number of colonies (4e, s4b, s4e, s5b), sometimes the % relative survival (6b), sometime the % normalized to the ratio of the treatment. The figure legends are equally variable and confusing. One consistent way to display the same type of data should be selected, and any change from it should be explained and well described in the figure legend. The rescue SB experiments are particularly inconsistently displayed and result a bit confusing (fig 6f vs. s5b vs. s5d). Another (minor) inconsistency is the relative positioning of the soft agar vs. colony survival assay panels, which are swapped frequently, including within the same figure (fig 6). **–All the clonogenic assays are now referred to as "Clonogenic assays". Fig. 4E is now Fig. 5E, Fig. 6B is now Fig.7B, Fig.6fF is now Fig. 7F, Supplementary Fig. 4B is now Supplementary Fig. 9B, Supplementary Fig. S4D- is now Supplementary Fig. 9D, Supplementary Fig. 5B is now**

Supplementary Fig. 10E, Supplementary Fig. 5D- is now Supplementary Fig. 10G .The Y-axis “% of survival after oligo treatment” in Fig. 6f, Supplementary Fig. 5D (now Fig. 7f, Supplementary Fig. 10G) was changed to “% Relative Survival”. In these cases the Y-axis is not “number of colonies” since these graphs are manipulated to show the relative survival of cells treated with oligonucleotide and SB compared to cells treated with SCRM and no SB. Fig. S5b (now Supplementary Fig. 10E) is the raw data of Fig. 6F (now Fig.7f). Y-axis in Fig. 6b (now Fig. 7b) is “% relative survival” and not “number of colonies” since it is a quantification of several assays (and not one). The actual number of colonies in each assay was different, for instance the number of colonies in SCRM could be 70 or 100 or 50, but the fold change compared to the oligonucleotide treatment was the same. Therefore, those experiments were quantified for one graph and a p-value presented.

"- Figure 7. There seem to be some inconsistency in the mCherry signal intensity of the SF212-treated tumors in panel 1 and 8 comparing the CY3 and BF/CY3 images **–We thank the review for pointing out this mistake which we have now fixed. Because of brightness problems (in the BF some pictures were too bright and the Cy3 was not seen well, we separated the exposures instead of overlaying them. We also removed mouse #9 (no tumor) from the SF2i2 treated group to make room for a diagram of the experimental design (panel A).**

Other points:

"- The sequences of the decoy oligos should be specified in the text of the article and/or in one of the main figures (maybe Fig 1a?). An alignment with the consensus motifs of the relative protein should be included (since some have degenerate/multiple binding motifs). As is, the sequences are only included in supplementary table 3 (with the RT-PCR primers), and the consensus are not presented." **–As the reviewer suggested we have now included in Fig 1a the consensus sequences of the splicing factor motifs and the sequences of the chosen decoys.**

"- The authors suggest (page 6, lines 3-5) that the discrepancy in the results between knock-down and functional inhibition by decoys may be accounted for by the RNA-binding-independent function of RBFOX1/2. This is a very intriguing hypothesis, which could be simply tested (at least initially) by verifying the presence of RBFOX1/2 binding sites in proximity of the splicing events affected in the two data sets." **–We agree with the reviewer. We analyzed the number of RBFOX1/2 motifs in the 250 nt upstream and downstream introns of each event. We now included in Fig. 2b, next to the Venn diagram, the number of consensus motifs found in the downstream introns of each group (Damianov study and our study). The results, which are statistically significant, show higher enrichment of RBFOX1/2 motifs in the downstream introns in our study compared to the knockout study. In the upstream introns the enrichment was smaller in both studies and the difference was not statistically significant.**

- Both in the introduction and in the discussion, the authors assert that because the decoys bind to the RNA-binding domain (true), they will only affect the RNA-binding/splicing activity of the splicing factor (likely but not proven), and thus they are more desirable than -for example- knock-down approaches, because they would have fewer side effects, thus presumably being more specific. However, it is unclear why the targeting of SFs should be in principle limited to their “RNA binding/splicing” activities, and/or why this should necessarily be an advantage, since some of their pathogenic functions could well involve their non-RNA-binding properties. In some cases the opposite might even be true. Furthermore, such different activities are frequently connected and it might be complex to decouple them. The decoy approach is different from knock down strategies, and it might prove to indeed lead to a superior therapeutic strategy for at least some of the targets, but it is not inherently ‘better’, based on the above argument, than knock-down. On the other hand,

some of the other arguments are convincing, such as that decoy might be more effective because they are more drug-like and do not depend on half-life and expression, that they can target a protein across species and can be delivered better. The discussion should be amended accordingly. **–We agree and now added to the discussion this paragraph (Page 18):** *“ Inhibition of pleotropic splicing factors such as PTBP1, SRSF1 or RBFOX1/2 is expected to affect the splicing of hundreds of genes ²⁵. Thus, the question of toxicity of their inhibitors is a major concern for such an approach. However, many oncogenes and oncogenic pathways (e.g. c-Myc, EGFR, Akt, mTORC1, Ras/Raf and others), which regulate normal growth, are valid targets for cancer therapy. The question of toxicity due to inhibition of each of these splicing factors and others must be evaluated carefully. Another complexity in inhibiting splicing factors is that they function in many steps of RNA processing; splicing, mRNA export, NMD and translation ⁵⁴. Thus, even a specific inhibition of their RNA binding activity can affect several different functions. Some of these functions may be beneficial for inhibition of their oncogenic activity and other may not. The advantage of competitive decoy oligonucleotides over knockdown methods is that the level of inhibition can be better controlled in a dose-dependent manner. In addition, the decoy oligonucleotides do not interfere with other functions of the splicing factors (not related to RNA processing), a fact that is expected to reduce the toxicity of this approach compared to knockdown of the splicing factor. “*

As far as specificity, whereas it is true that targeting just the RNA-binding activity might lead to “fewer side effects” than knocking down the protein tout-court as far as the intended target, it is also clear that the decoy could have much broader sequence-dependent off-target side effects, due to the redundant and degenerate binding motifs of SFs. For example, SF2i2 can clearly bind dozen of targets with apparent higher affinity than SRSF1 itself (fig S3a), which is likely not true for a well-designed siRNA. **–We agree and now modified the text (Page 17-18):** *“ In contrast to the RBFOX1/2 decoy oligonucleotides, which showed specific binding to RBFOX1/2 proteins in the proteomic analysis, the SRSF1 decoy oligonucleotides bound multiple proteins in addition to SRSF1. This lack of specificity could be attributed to the SR protein family which is known to have - redundant activity for some substrates in vitro and in vivo ^{52,53}. Another explanation is the limitation of the proteomic assay. This assay cannot distinguish between proteins that bind directly to the decoy oligonucleotides and indirect binders that bind SRSF1 through protein-protein interactions. A large number of proteins identified in the SRSF1 proteomic analysis have been recently demonstrated to bind SRSF1 through protein-protein interactions ⁴⁰. Even though the binding motif of some splicing factors is not well defined, we observed high specificity to specific splicing targets. SRSF1 targets were solely affected by the SRSF1 decoy oligonucleotides and not by the PTBP1 and RBFOX1/2 decoy oligonucleotides and vice versa, the SRSF1 decoy oligonucleotides did not affect splicing of some of the PTBP1 and RBFOX1/2 splicing targets (Supplementary Figures 8 and 10A-C). “*

"- To address specificity, some splicing events could be analyzed that are predicted NOT to be affected by the decoy, for example, RBFOX targets from Fig 1 could be analyzed in PTBP1i-treated and Sf2i2-treated cells, and vice-versa with PTBP1i targets from fig 4 and SF2i2 targets from fig 5." **–We agree with the reviewer. It is expected that some targets will be regulated/affected by multiple splicing factors while others will be more specific /depend on a specific splicing factor. As the reviewer suggested, we have now identified two splicing targets for each splicing factor (RBFOX2, PTBP1, SRSF1) which are affected only by the**

specific decoy oligonucleotide against the splicing factor (new Fig. S8). These results suggest that for some splicing targets, high specificity can be achieved by the decoy oligonucleotides. We now describe and explain this in the text (Page 12 lower part): " *To further evaluate the specificity of all decoy oligonucleotides, two splicing targets of each splicing factor were chosen; FMNL3 and NUMA1 for RBFOX1/2, RTN4 and SNAP91 for PTBP1 and USP8 and U2AF1 for SRSF1. Changes in splicing of each target were evaluated after introduction of each decoy oligonucleotide (Supplementary Fig. 8). Changes in splicing of FMNL3 and NUMA1 were seen only after transfection of RBFOX1 but not PTBP1 or SF2i2 (Supplementary Fig. 8A). Changes in splicing of RTN4 and SNAP91 were seen only after transfection with PTBP1 (Supplementary Fig. 8B) and significant changes in USP8 and U2AF1 splicing were observed only after transfection with SF2i2 but not with any of the other decoy oligonucleotides (Supplementary Fig. 8C). These results further strengthen our conclusion that each decoy oligonucleotide acts specifically on its targeted splicing factor. It is important to note that other splicing events may not behave in a similar manner to these specific splicing targets, since they might contain different cis-elements and be regulated by several different splicing factors at the same time.*"

"- Previous uses of decoy ASO strategies should be mentioned in the introduction even if (or especially because) they differ from the strategy proposed here." –We agree and now added a full paragraph in the introduction describing other oligonucleotide strategies (page 3): " *Current oligonucleotide technologies include: Antisense GAPmers, which are designed to knockdown gene expression by binding to specific mRNAs and activating their degradation by RNaseH ; Splice Switching Oligos which hybridize to pre-mRNA molecules, interfere with the binding of splicing factors or spliceosomal components and shift the splicing between splice sites or affect inclusion/skipping of specific exons and siRNAs which are designed to knockdown gene expression and are usually double stranded (reviewed in ⁶). All of these oligonucleotide technologies are based on binding/hybridizing to either pre-mRNA or mRNA. Here we present a novel technology using sense oligonucleotides that bind to RNA binding proteins rather than RNA. . The only known similar approaches of nucleic acids designed to bind proteins ^{7,8} are DNA (double stranded) oligonucleotides, that act as transcription factor decoys ^{7,8} and RNA aptamers, which are RNA molecules (sometimes much longer than the RNA oligonucleotides mentioned above) with a specific 3D structure that can bind many types of proteins according to the designed specificity (not only RNA binding proteins) ⁹. "*

"- Some space in the discussion should be dedicated to address the possible toxicity/on-target side effects of this approach, since splicing factors like SRSF1 have broad functions also in normal cells." –We agree and have now added to the text (bottom of page 17): " *Inhibition of pleiotropic splicing factors such as PTBP1, SRSF1 or RBFOX1/2 is expected to affect the splicing of hundreds of genes ²⁵. Thus, the question of toxicity of their inhibitors is a major concern for such an approach. However, many oncogenes and oncogenic pathways (e.g. c-Myc, EGFR, Akt, mTORC1, Ras/Raf and others), which regulate normal growth, are valid targets for cancer therapy. The question of toxicity due to inhibition of each of these splicing factors and others must be evaluated carefully. Another complexity in inhibiting splicing factors is that they function in many steps of RNA processing; splicing, mRNA export, NMD and translation ⁵⁴. Thus, even a specific inhibition of their RNA binding activity can affect several different functions. Some of these functions may be beneficial for inhibition of their oncogenic activity and other may not. The advantage of competitive decoy oligonucleotides over knockdown methods is that the level of inhibition can be better*

controlled in a dose-dependent manner. In addition, the decoy oligonucleotides do not interfere with other functions of the splicing factors (not related to RNA processing), a fact that is expected to reduce the toxicity of this approach compared to knockdown of the splicing factor.

"

- Figure legends should be fleshed out.

- In general, the text should be revised to improve the flow, as some parts still read as a draft and it seems overall a bit disorganized, particularly the introduction. For example, page 2, lines 3-4 is a verbatim repeat of page 1 lines 2-3; on page 3, line 10, the LASR complex should be referenced (ref 17) when it is defined; numerous typos are present throughout the manuscript. **-We hope the new version is more logic and flowing.**

Reviewer #2 (Remarks to the Author):

Splicing factors bind to specific RNA sequences. It is logical, therefore, that modified RNAs might be able to block splicing factors and selectively control their activities.

Comments

1) The premise of the paper is that RNAs that block splicing factors might be therapeutically important. This premise is weak. Compounds that block splicing factors would affect the splicing of many genes and lead to extensive off-target effects. Given that synthetic oligonucleotide complementary to specific gene targets struggle in the clinic with off-target effects, the notion that an therapeutic RNA might disrupt multiple gene expression events is not plausible. **-We agree with the reviewer and understand his pessimism. Many drugs and molecules fail on the way to the clinic. We agree that our proposed new technology still needs to be examined further before it can be developed to be a therapy. However, we still think it is a valid approach. As for the wide effects of inhibiting a splicing factor, this is true. However, inhibition of the cellular translation machinery by mTOR inhibition and inhibition of upstream growth factor receptors that regulate the activity of many signaling pathways and expression of hundreds of genes (EGFR, HER2, VEGFR) are all valid targets with approved drugs. Thus, it is of course a matter of general toxicity compared to anti-tumor activity. In cancer, as opposed to monogenic diseases, sometimes a strong hit to an essential cancer gene is more effective than a specific targeted therapy. Many of the drugs against the pathways I mentioned above are actually not toxic enough to cancer cells and thus only partially inhibit cancer. On the other hand, the reason chemotherapy (which is with no doubt toxic) still prevails is that it is effective and in some cancer types even a cure still today. Because there is no doubt that what the reviewer raised is a very important issue and one must be careful when presenting a new approach such as this we now added a paragraph to the discussion (bottom of page 17): *Inhibition of pleotropic splicing factors such as PTBP1, SRSF1 or RBFOX1/2 is expected to affect the splicing of hundreds of genes*²⁵. *Thus, the question of toxicity of their inhibitors is a major concern for such an approach. However, many oncogenes and oncogenic pathways (e.g. c-Myc, EGFR, Akt, mTORC1, Ras/Raf and others), which regulate normal growth, are valid targets for cancer therapy. The question of toxicity due to inhibition of each of these splicing factors and others must be evaluated carefully. Another complexity in inhibiting splicing factors is that they function in many steps of RNA processing; splicing, mRNA export, NMD and translation*⁵⁴. *Thus, even a specific inhibition of their RNA binding activity can affect several different functions. Some of these functions may be beneficial for inhibition of their oncogenic activity and other may not. The***

advantage of competitive decoy oligonucleotides over knockdown methods is that the level of inhibition can be better controlled in a dose-dependent manner. In addition, the decoy oligonucleotides do not interfere with other functions of the splicing factors (not related to RNA processing), a fact that is expected to reduce the toxicity of this approach compared to knockdown of the splicing factor.

2) This thrust to show that an anti-SF RNA might be a therapeutic adversely affects the structure of the paper by causing it to focus on weak in vivo experiments rather than rigorously making a case for the basic science hypothesis that anti-SF RNAs are possible. **–The reviewer’s comments (2-5) deal with a very important point that was indeed missing in the paper – a strong biochemical proof that justifies the approach, shows its biochemical basis and connects the basic biochemistry to the in vivo results. With the help of the biochemical expertise of Antoine Clery (laboratory of Fred Allain) in RNA-Protein interactions and the use of a new technology (switchSENSE) we were able to perform experiments that show:**

-A decoy oligonucleotide containing several motif repeats binds multiple protein molecules (the experiment is not fully quantitative so the exact number of molecules must be determined by other techniques). This result provides an explanation as to how an oligonucleotide with four repeats can inhibit several splicing factor molecules.

-As the number of repeats in the decoy oligonucleotide increases (1-4), the affinity to the splicing factor increases, with a much lower Kd for the decoy oligo with 4 repeats. This result suggests that there might be some cooperativity in the binding of the splicing factors and also correlates well with our results in cells showing that oligonucleotides with 1-3 repeats have weaker splicing effects than oligonucleotides with 4-5 repeats (Supplementary Fig. 7b).

-The RBFOX RRM binds very specifically to the RBFOX decoy motif but not to a sequence similar to the SRSF1 binding motif, as determined by NMR.

These results are all in new Fig. 3, new Supplementary Fig. 4 and new Supplementary Fig 5.

3) If well-supported, the basic science showing that anti-SF RNAs are functional in cells would be suitable for well regarded journal like Nature Communications. **–We thank the reviewer for his/her comment and as mentioned above now provide the biochemical basis for the decoy technology.**

4) This paper supplies a wide array of data that are consistent with their RNAs having the desired effects. However, much of the data is superficial and unpersuasive. In this reviewers opinion, a much more tightly focused paper focusing on supporting the fundamental hypothesis would have been preferred. **–As mentioned above, we have now strengthened the biochemical explanation to the fundamental hypothesis.**

5) There are no biochemical experiments exploring the binding potency and specificity of SFs with RNAs. The hypothesis that a much longer than native 2'-O-methyl RNA can bind specifically to an SF is a remarkable one and requires beginning with simple biochemical assays. **–We agree, as mentioned above. We think the reviewer’s suggestions and the experiments we have done to answer his questions have strengthened the basis for the decoy technology. We now provide a biochemical explanation to phenotypes we observed in cells (e.g. the stronger effects of decoy oligonucleotides with more than 3 repeats compared to oligonucleotides with**

6) Western analysis of pull down experiments is of limited value. One would expect that, at a high enough concentration, an RNA binding protein will bind its cognate sequence. **–We agree.**

7) Mass spectrometry is not a trivial experimental technique. It is prone to experimenter bias and cherry-picking. The experimental description of the mass spectral analysis is entirely deficient, making it impossible to judge the rigor of the experiments. **–We apologize for the unclear description of the MS experiments and now provide a more elaborate explanation (see also above the answer to reviewer 1). We also apologize for the MS tables that were not uploaded properly, preventing the reviewers from examining them. We hope now the reviewers will be able to see all the data and the explanations.**

8) It is not clear why an oligonucleotide with three or four binding motifs would associate with a splicing factor more strongly than an oligonucleotide with just one motif. The splicing factor can bind just one motif at a time. While I do not exclude that it is conceivable that the design concept might be viable, the lack of precedent is a good example of why a much more careful biochemical framework is essential. There is no recognition that adding binding sites to increase affinity will seem illogical to many and needs to be carefully defended. **–The reviewer’s question echoed in our minds and drove us to perform the switchSENSE experiments with all the oligonucleotides containing 1-4 repeats and to design the experiment that measures how many protein molecules bind to oligonucleotides containing 1, 2, 3 or 4 repeats. So now there is an answer to this basic and important question: Indeed, the affinity of an oligonucleotide with 4 repeats is 5 times higher than an oligonucleotide with 1 repeat. The reason is probably the fact that an oligonucleotide with 4 repeats can bind several (we still don’t know the exact number) protein molecules of the splicing factor. There might also be cooperativity in the binding that might explain the Kd drop when reaching 4 repeats.**

9) For experiments that examine anti-growth phenotypes upon addition of oligonucleotides, more controls are necessary. Many oligonucleotides have off-target anti-proliferative effects. This is a problem for both Figures 6 and 7. For Figure 7, oligos are notorious for having anti-tumor effects that are unrelated to binding to their intended targets inside cells. **–To address the reviewer’s concerns we have repeated several oncogenic assays with 3 additional oligonucleotides with identical chemistry that were originally designed in our lab to act as antisense oligonucleotides to exon 78 of the dystrophin gene, which is expressed mostly in muscle cells. These chosen ASOs were not effective in modulating exon 78 splicing. The new data which is presented in new Supplementary Fig. 10A-C shows that these 3 oligonucleotides did not significantly affect the oncogenic properties of U87MG cells compared to the SF2i2 decoy oligonucleotide that was tested in the same experiment.**

10) How many copies of the decoy enter cells, are released from endosomes, and enter the nucleus? How many copies of splicing factors are there? Is the decoy effect plausible from the standpoint of molecular stoichiometry? **–Even though all of the experiments in this paper (except for the zebrafish experiments which used a direct injection) were done by transfecting the decoy oligonucleotides using lipofectamine 2000 which has been used for many years by many labs and for many different oligonucleotides, we performed the suggested experiments using high resolution microscopy and live imaging measuring endosomal uptake by labeled dextran. We do not know if the results are surprising as this is not free uptake of the oligonucleotides but rather regular lipofectamine transfection (which is highly efficient world-wide). Most of the labeled decoy oligonucleotide (Cy5) did not co-localize with endosomes (FITC). Within 1-2 hours there were already oligonucleotides inside the cells (new Supplementary Fig 2, new Supplementary movies). After 11 hours at least 20-30% of the oligonucleotides were in the nucleus (new Supplementary Fig. 3, new Supplementary Table 1). We believe that after longer time periods additional oligonucleotide**

molecules can still enter into the nucleus. Importantly, splicing effects of the oligonucleotides are detected as early as 4-8 hours post transfection (Fig. 2a).

11) Oligos with PS backbones tend to be nonspecifically bind proteins, increasing the requirement for rigorous controls and cautious interpretation of experiments. **–In this paper, only the zebrafish experiments were performed with oligonucleotides with a PS backbone.**

12) This review should not be construed as a complete list of the problems with this paper. The entire approach needs to be more carefully thought out from start to finish. **–We have now integrated the new biochemical experiments into the paper and added a paragraph in the introduction describing the different oligonucleotide technologies as well as decoy approaches for other proteins and discuss the pros and cons of the strategy including the possible toxicity issues. We have elaborated our explanations of some of the assays and believe that with the new experimental data and text modification the paper is much improved.**

Remarkable results demand remarkable proof. A short, repetitive oligonucleotide that binds selectively to a splicing factor and acts as a decoy would be a remarkable result. The authors have fallen into the trap of putting together a paper that touches on many experiments but does none of them convincingly. complex experiments are presented with a gross lack of clear supporting data. Controls are minimal. A paper that focuses on only validating the basic hypothesis and does so convincingly would advance the field and be a better candidate for publication. **–We believe the additional experiments and textual modifications we have incorporated into the manuscript to address the reviewer’s questions and comments have made our conclusions more solid and improved the manuscript compared to the previous version. We hope the reviewer will now find it suitable for publication in Nature Communications.**

We thank both reviewers for their comments and questions. We think that after addressing these comments the paper is now much more solid both biochemically and biologically.

We hope the reviewers will find it now suitable for publication in Nature Communication.

With best regards,

Rotem

Reviewers' comments:

Reviewer #1 (Remarks to the Author):

Denichenko_2018_NComms

In the resubmission of their manuscript "Specific Inhibition Of Splicing Factor Activity By Decoy RNA Oligonucleotides", Denichenko and colleagues thoroughly address all the major concerns of this reviewer, which is now in my view acceptable for publication.

The current version of the manuscript is significantly more complete and reads much better than the previous version. The switchsense suite of experiments behind Figure 3 adds mechanistic understanding and is a welcome addition. The presentation of the data in the tables and supplementary material is largely improved and satisfactory.

Some minor issues should be addressed, which don't impact the enthusiasm for this work:

- line 108. Knock-down of SF would have more pleiotropic effects on the cells. Whether these would be "negative", would have to be shown experimentally, especially in a therapeutic context. Text should be modified accordingly.

- line 109. Similarly, it's not clear why "ideally", a decoy ASO should only target splicing activity. As an investigational tool, I would agree, as it gives a cleaner picture. As a therapeutic lead, not necessarily. We just don't know.

- line 131. The SCRM sequence should be included in Fig 1a.

- line 217. When discussing the possibility of cooperative RNA binding by RBFOX1/2, the authors should mention and reference some of the rich literature in this field, for example reviewed by Dassi, Front Mol Biosci. 2017; 4: 67.

- line 353. Referral to "(Fig. 8a-c)" is inaccurate. Either refer to (Fig. 8 b, d) in line 352 and (Fig. 8 a, c) in line 353, or just (Fig. 8) in line 353.

- line 366. If splicing factors have multiple physiological functions, including some not depending on their RNA-binding properties, the removal of these function by knock-down should not be regarded as "side effects", as they are proper effects. Thus "fewer side effects" should be substitute by "more splicing-specific effects" or something along those lines.

- lines 371-374. If space is needed, this paragraph could be omitted. It is an interesting observation, but largely immaterial to the rest of the work, especially considering there are plenty of non-RNAi methods to knock down proteins.

-line 424. "presents the first type of intelligence-based design..." seems a bit excessive. Something along the lines of "introduces a new class of molecules designed to directly inhibit..." would be preferable.

Reviewer #2 (Remarks to the Author):

Sadly, the fundamental rethinking of how this work is approached has not occurred.

1. The microscopy for oligonucleotide uptake uses fixed cells. It has been know for almost fifteen

years that fixation liberates tagged oligo from endosomes and leads to misleading conclusions about localization.

2) Table 2 appears to show binding of many proteins. In any event, the table is impossible for this reviewer to interpret. Mass spec is inherently prone to cherry-picking because of its sensitivity. The burden is on authors to make a case for data reliability. That case was not made here.

3) I was unable to locate the sequence of the control oligonucleotides they are not listed in Table S5. That omission (assuming they are not somewhere I have yet to locate) is disqualifying.

4. The scrambled control is, in fact, not a scrambled control. The ratio of bases does not mimic the "active" compound

5. As I stated in my previous review, the manuscript needs to be entirely rethought and refocused. One negative control is entirely insufficient for experiments that break novel ground like this one (and the design of the control was flawed)

6. The reported binding difference between having one and four binding sites is probably too small to have a biologically significant impact.

7) Too few controls are used for the fish experiment.

8) Too few controls are used in Figures 5 and 6

9. Survival assays often produce misleading results. Perhaps the SCRM oligo (which was not well chosen) has characteristics that reduce its toxicity?

10. Splicing changes, not luminescence changes, should be shown in Figure 8.

Any new addition to the text we made in this version of the manuscript is marked in red.

Reviewer #2 (Remarks to the Author):

Sadly, the fundamental rethinking of how this work is approached has not occurred. **-We have added a new section regarding the binding affinity and binding stoichiometry of decoy oligonucleotides (new Fig. 3, new supplementary Figs. 3-4) which deepens our understanding of how the decoy oligos bind to their protein targets. Unfortunately, the reviewer ignored the new figures that address the major comments of this reviewer in his first review.**

1. The microscopy for oligonucleotide uptake uses fixed cells. It has been known for almost fifteen years that fixation liberates tagged oligo from endosomes and leads to misleading conclusions about localization. **-All the staining showing the kinetics of labeled decoy oligo entry into the cells, labeling of endosomes and entry into the nucleus were performed on live cells, including a movie that shows the kinetics (Supplementary Figs. 2, 3 ;Supplementary Movie 1). Text is on page 6 and Supplementary figure legends.**

2) Table 2 appears to show binding of many proteins. In any event, the table is impossible to for this reviewer to interpret. Mass spec is inherently prone to cherry-picking because of its sensitivity. The burden is on authors to make a case for data reliability. That case was not made here. **-In the revised manuscript, we included a new and detailed explanation of the proteomic analysis, as requested also by reviewer #1, and reorganized the tables as reviewer #1 suggested, to his full satisfaction. There is no "cherry-picking" as we present all the data, including the non-specific binders and it is clear that some oligos bind multiple proteins with lower specificity.**

3) I was unable to locate the sequence of the control oligonucleotides they are not listed in Table S5. That omission (assuming they are not somewhere I have yet to locate) is disqualifying. **- We apologize for overlooking the inclusion of the sequences of the three new control oligos. These sequences are now added in the revised version (Table S5).**

4. The scrambled control is, in fact, not a scrambled control. The ratio of bases does not mimic the "active" compound **-This is a new claim, which was not raised previously. As this is not an antisense technology, the meaning of base composition is a bit less relevant. The scrambled oligonucleotide was designed similarly to the SRSF1 decoy but with disruption of the SRSF1 binding consensus sequence. We included in the revised manuscript (per the request of reviewer #2) three new control oligos with various base compositions (Supplementary Figs. 9 and 11, Table S5). We performed cross analysis of the effects of each decoy oligo on the targets of the other decoy oligos (Supplementary Fig. 8). In addition, we determined the specificity of the splicing signaling and biological effects of the SRSF1 decoy oligo compared to the three new control oligos, showing that the control oligos do not show the same biological effects (Supplementary Figs. 9-11).**

5. As I stated in my previous review, the manuscript needs to be entirely rethought and refocused. One negative control is entirely insufficient for experiments that break novel ground like this one (and the design of the control was flawed). **-As stated above, per the request of the reviewer in the previous round of review, we included three new control oligos and showed that they do not affect any oncogenic property of the cells, while the SRSF1 decoy does**

(Supplementary Fig. 11). Moreover, we now show that the three new control oligos do not affect the splicing of SRSF1 targets and do not activate the p38-MAPK pathway, while the SRSF1 decoy oligo does (Supplementary Fig. 9). Furthermore, we added in the previously revised version of the manuscript cross-target validation showing the splicing effect specificity of each decoy oligo (Supplementary Fig. 8). Our initial experiments with SRSF1 decoy, RBFOX1/2 decoy and scrambled oligos were performed side by side. We did not include the negative results of the oncogenic effects of the RBFOX1/2 decoy in the previous versions of the manuscript. Although the RBFOX1/2 decoy strongly affects dozens of splicing events (Figures 1,2,4) it had no effect on the oncogenic properties of the cells. We now include this data in Supplementary Fig. 10.

6. The reported binding difference between having one and four binding sites is probably too small to have a biologically significant impact. **–The basis of this statement is unclear; the data, however, is very clear. The binding difference between one and four repeats is more than fourfold (in vitro, with a recombinant protein – in vivo this might be even stronger). This by itself is biologically significant as was already observed by Fred Allain’s group showing a strong effect on splicing regulation with mutant proteins for which the mutation decreased their affinity by a factor of five, in vitro (Clery et al., 2011 NSMB ; Clery et al., 2013 PNAS). We now added this statement to the text (page 17): " It has previously been shown that mutant splicing factors, for which the mutation decreased their affinity to the target RNA motif in vitro by a factor of five, had reduced splicing activity in cells 42,52."**

7) Too few controls are used for the fish experiment.

The experiments in Fig. 4 closely follow the experiments described in Gallagher et al. (2011), where zebrafish were injected with morpholino antisense oligos that silence RBFOX1/2. All the alternative splicing events and phenotype observed in the muscles were detected with a completely different oligo, different chemistry and different mechanism (knockdown versus decoy in our case). Raising this comment at this stage in the review process, gives us the impression that the reviewer is not acting in good faith.

8) Too few controls are used in Figures 5 and 6

–These two points repeat point 5. We included in the previously revised manuscript experiments on the oncogenic effects of three new control oligos. We now added experiments showing the lack of splicing effects and changes in phosphorylation state of p38-MAPK caused by these three new control oligos (related to experiments in Figs 5-6) (new supplementary Fig. 9). We also now added experiments showing no oncogenic effects of the RBFOX1/2 decoy. Finally, the reviewer ignored the cross-target validation we performed. The amount of controls in this paper is very high using a range of assays and cells with four oligos (apart from the scrambled oligo) that do not show any oncogenic effect (Oligo Cont. 1,2,3, RBFOX1/2 decoy), three oligos that do not have splicing effects (Oligo Cont. 1,2,3). We demonstrate that only one oligo affects the activation of the p38-MAPK pathway (SRSF1 decoy, Supplementary Fig. 9). Every experiment in this paper was repeated 6-7 times and the data is highly reproducible. Raising these points at this stage in the review process, being aware that we already performed additional experiments with three control oligos, gives the impression that the reviewer is not acting in good faith.

9. Survival assays often produce misleading results. Perhaps the SCRM oligo (which was not well chosen) has characteristics that reduce its toxicity? **–According to the request of the reviewer, we included in the previously revised version three new control oligos and show that they do**

not affect colony survival (clonogenic assay), soft agar colony formation (anchorage independent growth) and proliferation (Supplementary Fig. 10). Unfortunately, the reviewer ignored these results or overlooked them in the revised version.

10. Splicing changes, not luminescence changes, should be shown in Figure 8. –The reviewer is raising a new point, which he did not raise previously. Moreover, in these experiments the cells were transfected with the decoy oligonucleotides, prior to injection into the mice, similarly to other experiments shown in the paper. Therefore, splicing changes in similar experiments can be seen in Fig. 6 and Supplementary Figs. 7-8. This experiment represents only the effects of the SRSF1 decoys following transfection into cells and then injection of the cells in vivo and is not meant to represent any type of treatment. There is an explanation in the text and we do not claim that this is a model system for treatment.

REVIEWERS' COMMENTS:

Reviewer #3 (Remarks to the Author):

In this study, Denichenko and co-authors focus on the use of decoy oligonucleotides, composed of several repeats of an RNA motif, to inhibit the activity of individual splicing factors (SFs). This approach was used to target RBFOX1/2, SRSF1 and PTBP1.

Through a series of experiments, which include adequate controls, the authors convincingly show that these decoy oligonucleotides do bind to their respective SFs with specificity and inhibit their splicing activity both in vitro and in vivo.

The authors present a large number of experiments ranging from AS analysis to determining phenotypes in zebrafish, to assessing the oncogenicity of cell lines. The amount of work is impressive and the quality of the data is very high. The contribution of the Allain lab is evident with important validation data on the binding of decoy oligonucleotides to the respective SFs.

The authors may wish to tone down the therapeutic implications and at the same time emphasise the clear advantage of using decoy oligonucleotides to inhibit the activity of RBPs, over more drastic knock-down approaches.

In summary, I think that this is an excellent paper, which establishes the proof of principle of using decoy oligonucleotides to inhibit the activity of SFs with specificity.

Response to referee requests

Reviewer #3:

"The authors may wish to tone down the therapeutic implications and at the same time emphasise the clear advantage of using decoy oligonucleotides to inhibit the activity of RBPs, over more drastic knock-down approaches."

-To answer reviewer #3 we changed the end of the abstract:

"These decoy oligonucleotides present an approach to specifically downregulate SF activity ~~and have the potential to treat diseases in conditions where SFs are either up-regulated or hyperactive such as cancer.~~"

We also deleted the last sentence of the discussion, which is the only place in the paper where the wording "therapy" refers to the decoy oligonucleotides and we added the fact that this technology includes RNA binding proteins (RBPs):

"In conclusion, the decoy oligonucleotide technology presented here is the first type of intelligence-based design of molecules that can directly **and specifically** inhibit the activity of ~~splicing factors~~**RNA binding proteins (RBPs), such as splicing factors**; a substantial advantage over traditional knockdown methods.~~This technology has the potential to be developed as a therapeutic approach to treat diseases known to involve overexpression or high activity of specific splicing factors.~~"